# A novel rolling bearing fault detect method based on feature mode decomposition and subtraction-average-based optimizer

**Wei Xi**¤, **Fuyu Qiao**, **Jingjing Zhang***

School of Electrical Engineering, Hebei University of Architecture, Zhangjiakou, Hebei, China

¤ Current Address: School of Electrical Engineering, Hebei University of Architecture, Zhangjiakou, Hebei Province China
* zhangjeileen@163.com

## Abstract

Large rotating machinery is an essential piece of equipment in modern industry, playing a critical role in industrial production. However, the complex working environment complicates the extraction of fault-related information. This paper proposes a fault diagnosis method based on the subtraction-average-based optimizer (SABO) and feature mode decomposition (FMD). To address the issue that FMD's decomposition performance is highly sensitive to its parameter settings, this paper uses the minimum envelope entropy as the fitness function and employs the SABO algorithm to adaptively optimize FMD's two key parameters: the mode number (n) and filter length (L). Additionally, for the intrinsic mode functions (IMFs) obtained from FMD decomposition, the maximum kurtosis value is used to filter IMFs containing fault information, and envelope spectrum analysis is applied to achieve fault diagnosis. When applied to experimental signals of rolling bearing faults, the results demonstrate that the proposed method can extract the amplitude of the fault characteristic frequency from the envelope spectrum and accurately diagnose the fault type. Compared with methods based on empirical mode decomposition (EMD) and fixed-parameter FMD, the proposed method provides a more prominent representation of the fault characteristic frequency and its harmonics in the envelope spectrum. Furthermore, the proposed method achieves a more prominent representation of the fault eigenfrequency in the envelope spectrum and a lower error rate. The proposed method demonstrates significant potential and value for rolling bearing fault diagnosis.










## 1 Introduction

As an important production equipment in modern industry, large rotating machinery often assumes a key role in aerospace, construction, railway, and other domains, and its failure will lead to huge downtime losses [1,2]. Research has shown that bearing

**Data availability statement:** All relevant data are within the manuscript and its Supporting Information files.

**Funding:** This research was funded by basic scientific research business funds of colleges and universities in Hebei Province (Grant No. 2023JCTD05). it in fact came from a project awarded to the first author of the paper (project number: 2023JCTD05). The category of this project is basic scientific research business funds of colleges and universities in Hebei Province.

**Competing interests:** The authors have declared that no competing interests exist.

failures account for 45–55 percent of all rotating machinery failures [3]. It is characterized by a complex working environment and a sophisticated dynamic system. Rolling bearing failures are mostly caused by cracking or spalling of the inner or outer race and wear or deformation of the rolling elements. Hence, it is essential to diagnose bearing faults simply and efficiently [4,5]. However, fault features are difficult to detect in fault signals due to heavy noise and interference from complex transmission paths. Therefore, modern information technology and signal processing methods are key to extracting weak fault features [6,7].

When the inner ring, outer ring, or rolling elements of the bearing are worn or damaged, a series of periodic pulses will be generated in the measured signal [8]. Therefore, periodic pulses in the vibration signal can serve as important indicators of mechanical failure. In general, the core of feature extraction is to extract periodic pulses from complex signals through matching and filtering techniques. Commonly used methods include spectral kurtosis [9], sparse filtering [10] and deconvolution [11]. However, as modern equipment evolves towards high integration, its structure and composition become increasingly complex, resulting in more components and interference in the measured signal. This significantly reduces the signal-to-noise ratio and hampers the extraction of fault information. As a result, traditional feature extraction methods are increasingly challenged. The fault decomposition approach separates various components of the original signal into several regular and simple modes, which can be easily analyzed in both the time and frequency domains. These approaches are considered among the most efficient tools for multi-component signal analysis.

In 1998, Huang et al. proposed empirical mode decomposition (EMD) [12]. In this method, the signal is decomposed into one or more intrinsic mode functions (IMFs). It has the advantage of not requiring a choice of basis functions. The decomposition of the signal depends entirely on the distribution of extreme points in the signal itself. However, many scholars have found that EMD is prone to mode mixing, both in the envelope and beyond, due to its unreasonable convergence conditions. To overcome these shortcomings, researchers have proposed improved methods.

In 2005, Smith proposed local mean decomposition (LMD) [13]. This method addresses the fitting problem of EMD, avoids the over-envelope defect, and mitigates the effects of mode mixing and endpoint effects. However, LMD remains a recursive modal decomposition method, and errors still occur during the decomposition process.

In 2009, Wu and Huang proposed ensemble empirical mode decomposition (EEMD) [14]. EEMD is effective for analyzing and processing nonlinear and non-stationary signals, and it addresses the mode aliasing problem of EMD during signal decomposition.

In 2010, Jia-Rong Yeh and Jiann-Shing Shieh proposed complementary ensemble empirical mode decomposition (CEEMD) [15]. CEEMD is an improved algorithm for noise-assisted data analysis. In this method, the residual of added white noise can be extracted from the mixture of data and white noise using pairs of complementary ensemble IMFs with positive and negative white noise. Although this approach yields

IMFs with a similar RMS noise level as EEMD, it effectively eliminates residual noise in the IMFs. However, a disadvantage of CEEMD is that redundant IMF components are generated if the amplitude of added white noise and the number of iterations is not appropriately chosen. Therefore, the IMF components need to be recombined or processed.

In 2011, M. E. Torres et al. proposed complete ensemble empirical mode decomposition with adaptive noise (CEEMDAN) [16]. This method has the advantage of requiring less than half the screening iterations of EEMD, and the original signal can be accurately reconstructed by summing the modes. As a result, a smaller ensemble size is required, reducing computational costs. However, this method also has some weaknesses, such as high algorithmic complexity, difficulty in parameter adjustment, sensitivity to noise, and mode mixing issues.

In 2013, Gilles put forward the empirical wavelet transform (EWT) based on EMD and the wavelet analysis method [17]. It overcomes the problems of EMD, such as incomplete theoretical foundations, the presence of modal aliasing, and how to accurately select the parent wavelet in wavelet analysis methods. However, the decomposition results of the method rely on the segmentation of the Fourier spectrum heavily.

In 2014, Dragomiretskiy and Zosso proposed variational mode decomposition (VMD), a new adaptive non-recursive decomposition method [18]. VMD has a stronger mathematical foundation than EMD and effectively reduces mode aliasing and endpoint effects. However, the decomposition efficiency of VMD depends on several parameters. Manual selection of parameters may degrade the decomposition efficiency of variational mode decomposition (VMD). Many researchers have proposed solutions to address this limitation.

For example, Diao et al. proposed an improved variational mode decomposition (VMD) method based on particle swarm optimization (PSO) [19]. The method combines PSO with a maximum entropy (ME)-based fitness function to optimize the penalty term and the number of modes in VMD. Support vector machine (SVM) is then used for pattern recognition. Experimental results demonstrate the effectiveness of the method. This method improves fault feature extraction accuracy by optimizing the VMD decomposition process.

Wang et al. proposed a rolling bearing fault diagnosis method based on whale optimization algorithm (WOA)-optimized VMD and graph attention networks (GAT) [20]. The method uses WOA to adaptively determine the penalty factor and decomposition level in VMD. The optimal parameter combination is then determined and used in VMD to decompose the original signal. Next, IMFs with high correlation to the original signal are selected using the Pearson correlation coefficient, and the selected IMFs are reconstructed to remove noise from the original signal. Finally, the graph structure data is constructed using the k-nearest neighbors (KNN) algorithm. Experimental results demonstrate that the method significantly improves the accuracy and robustness of rolling bearing fault diagnosis through optimized VMD parameter selection.

Yan and Jia proposed a fault diagnosis method based on cuckoo search algorithm (CSA)-optimized VMD and optimal scale morphological slice bispectrum (OSMSB) [21]. The method uses CSA to optimize VMD parameters, decomposes the bearing vibration signals into a series of IMFs using the optimized VMD, and selects the IMF with the highest fault feature ratio (FFR) as the principal component containing the most significant fault characteristics. OSMSB is then applied to enhance the fault features. Experimental results demonstrate that the method significantly improves the detection capability for rolling bearing outer ring faults through optimized VMD parameter selection. Based on the experimental results discussed above, we conclude that optimizing key VMD parameters using appropriate optimization algorithms can enhance the decomposition efficiency of VMD, thereby eliminating the impact of manual parameter selection on its performance.

In 2023, Miao et al. proposed feature mode decomposition (FMD) [22]. FMD is a new adaptive decomposition method. Briefly, FMD uses the correlation cliff of deconvolution as the objective function. It constructs an adaptive finite impulse response (FIR) filter bank and combines it with a correlation function to achieve fault period discrimination, ultimately decomposing the signal into a sum of multiple modal components. Unlike other signal decomposition methods, FMD fully considers the periodic and shock characteristics of the signal, ensuring sensitivity to fault characteristics while maintaining robustness to interference and noise.

However, similar to the VMD approach, the decomposition performance of FMD depends on two key parameters: the number of modes (n) and the length of the FIR filter (L). If these parameters are not properly selected, the decomposition performance of FMD will be significantly degraded. Many scholars have proposed solutions to address the challenge of parameter selection in FMD. For example, Lei et al. proposed a method to optimize FMD parameters using pelican optimization algorithm (POA) [23]. The method uses the index of all crags as the fitness function, optimizes FMD parameters using the POA algorithm, and combines envelope spectrum analysis to achieve fault diagnosis. Experimental results demonstrate that the method can extract the amplitude of the fault characteristic frequency and its harmonics from the envelope spectrum by optimizing FMD parameters, enabling the diagnosis of early-stage rolling bearing faults.

Li et al. proposed an adaptive parameterized FMD method [24]. The method employs a cuckoo search algorithm with logarithmic decline of nonlinear inertial weights and random adjustment discovery probability (DWCS) to optimize FMD parameters. The optimal parameter combinations are adaptively selected using DWCS based on the maximum feature frequency ratio (FFR).

Lv et al. proposed a high-performance rolling bearing fault diagnosis method based on adaptive FMD and Transformer [25]. The method uses the coati optimization algorithm (COA) to optimize two key FMD parameters (number of modes and filter length). It also incorporates Tent chaotic mapping, dynamic refractive inverse learning, and Levy flight strategies to enhance the algorithm's convergence speed and global search capability. A new fault diagnosis framework is established by integrating Transformer for global feature extraction, improving the accuracy and robustness of fault diagnosis. Based on the literature, a suitable optimization algorithm can assist FMD in parameter selection, effectively eliminating the impact of manual parameter selection and improving FMD's decomposition efficiency.

In 2023, Trojovsky Pavel and M. Dehghani proposed the subtraction-average-based optimizer (SABO), an optimization algorithm based on subtractive averaging [26]. SABO is a mathematically inspired optimization algorithm. It updates the positions of group members in the search space through subtractive averaging. SABO demonstrates high optimization capability and fast convergence. Experimental results demonstrate that SABO outperforms traditional optimization algorithms in terms of convergence speed, global search capability, and optimization accuracy. Therefore, SABO can assist FMD in selecting optimal parameters and improving its decomposition efficiency.

Based on this background, this paper proposes a new bearing fault diagnosis method based on SABO and FMD.

## 2 Method

### 2.1 Frequency modulation decomposition

Unlike conventional methods, FMD is a non-recursive decomposition approach. FMD initializes the FIR filter bank and updates the filter coefficients while adaptively selecting different modes. The decomposition process of FMD can be divided into three main steps: (1) initialization of the adaptive FIR filter bank, (2) adjustment of the filter bank, and (3) mode selection.

(1) Initialization of the adaptive FIR filter bank

In FMD, correlated kurtosis (CK) [27] is used as the objective function to update the filter. The initialization process involves applying a Hanning window based on the cut-off frequency. To implement the Hanning window initialization, the frequency band of the raw signal is first divided into K uniform segments. Then, the upper and lower cut-off frequencies of these segments, denoted as $f_u$ and $f_l$, are determined by equation (1).

$$\begin{cases} f_l = \frac{k \cdot f_s}{2K} \\ f_u = \frac{(k+1) \cdot f_s}{2K} \end{cases}, k = 0, 1, 2 \cdots K-1. \tag{1}$$

Here, $f_s$ represents the sampling frequency of the raw signal. After initialization, the filter coefficients are updated through continuous iteration to obtain an adaptive FIR filter bank.

(2) Adjustment of the filter bank

The FMD theory solves a constrained problem, as presented in equation (2).

$$\arg\max\left\{CK_M(u_k)=\frac{\sum_{n=1}^{N}\left(\prod_{m=0}^{M}u_k(n-mT_s)\right)^2}{\left(\sum_{n=1}^{N}u_k(n)^2\right)^{M+1}}\right\},u_k(n)=\sum_{l=1}^{L}f_k(l)\,x(n-l+1) \tag{2}$$

Here, $x(n)$ represents the original signal. L is the length of $x(n)$. $u_k(n)$ is the k-th decomposed mode. $f_k$ is the k-th FIR filter and its length is L. $T_s$ is the input period estimated using the Hilbert transform and autocorrelation spectrum (which is iteratively updated to approach the fault period), and M is the shift order. During FMD decomposition, an iterative eigenvalue decomposition algorithm is used to decompose the signal, and the decomposition mode can be expressed in matrix form as equation (3).

$$u_k=Xf_k \tag{3}$$

Here, $u_k=\begin{bmatrix}u_k[1]\\ \vdots\\ u_k[N-L+1]\end{bmatrix}$, $X=\begin{bmatrix}x(1) & \cdots & x(L)\\ \vdots & \ddots & \vdots\\ x(N-L+1) & \cdots & x(N)\end{bmatrix}$,$f_k=\begin{bmatrix}f_k(1)\\ \vdots\\ f_k(L)\end{bmatrix}$, The definition of CK for the decomposition mode is given in equation(4).

$$CK_M(u_k)=\frac{u_k^H W_M u_k}{u_k^H u_k} \tag{4}$$

Here, the superscripts H denotes the conjugate transpose operation. $W_M$ is an intermediate variable used to control the weighted correlation matrix. Substituting equation (3) into equation (4) yields equation (5).

$$CK_M(u_k)=\frac{f_k^H X^H W_M X f_k}{f_k^H X^H X f_k}=\frac{f_k^H R_{XWX} f_k}{f_k^H R_{XX} f_k} \tag{5}$$

Here, $R_{XWX}$ and $R_{XX}$ are the weighted correlation matrix and correlation matrix, respectively. Theoretically, maximizing equation (5)with respect to the filter coefficients is equivalent to finding the eigenvector associated with the maximum eigenvalue λ of the generalized eigenvalue problem in equation (6).

$$R_{XWX}f_k=R_{XX}f_k\lambda \tag{6}$$

Through iteration, the k-th filter coefficient is updated using the solution of equation (6) to approach the objective of obtaining a filtered signal with maximum CK. The accurate input period $T_s$ plays a critical role in updating the filter coefficients.

According to self-correlation theory, the self-correlation spectrum of the signal exhibits local maxima at period positions. If $R_x(\tau)$ represents the self-correlation value of signal $x(n)$, then $R_x(\tau)$ as a function of the of the lag τ is defined by equation (7).

$$R_x(\tau)=\int_{n=1}^{N}x(n)x(n+\tau)\,dn \tag{7}$$

The $\tau_0$ point in $R_x(\tau_0)=0$ is the zero-crossing point, and the estimated period of each filtered signal is chosen as the point at which the spectrum of self-correlation achieves a local peak $R_x(\tau_1)$ after passing through the zero-crossing point under condition $T_s=\tau_1$.

(3) Mode selection

Theoretically, the final modes contain specific components of the raw signal. FMD selects only the modes with maximum CK value. If all the initialized filters are updated throughout the FMD process, multiple modes may contain identical components. To eliminate mode mixing or redundancy, the mode with the largest correlation coefficient is initially locked, as a higher correlation coefficient indicates that the two modes share more identical components [28]. Meanwhile, to retain the modes with more fault information, the mode with the smaller CK value is discarded from the pair of modes with the maximum correlation coefficient. The correlation coefficient between two modes, $u_p$ and $u_q$ is defined by equation (8).

$$CC_{pq} = \frac{\sum_{n=1}^{N} (u_p(n) - \overline{u}_p)(u_q(n) - \overline{u}_q)}{\sqrt{\sum_{n=1}^{N} (u_p(n) - \overline{u}_p)^2}\sqrt{\sum_{n=1}^{N} (u_q(n) - \overline{u}_q)^2}} \tag{8}$$

Here, $u_p$ and $u_q$ represent the p-th and q-th decomposition modes, p and q are integers within the interval [1, 2, …, K]. $\overline{u}_p$ and $\overline{u}_q$ are the mean values of modes $u_p$ and $u_q$.

FMD uses CK as the objective function to update the filter. By setting the filter length (L) and the number of filters (K), an adaptive FIR filter is designed, and its coefficients are iteratively updated so that the filtered signal continuously approximates the objective function. Next, by setting the number of modes (n), the correlation coefficient (CC) between every pair of modes is computed to construct a K × K matrix. The two modes with the highest CC value are selected, and their CK values are computed using the estimated period. The mode with the lower CK value is then discarded. The remaining modes are retained as the final decomposed modes.

### 2.2 Subtraction-average-based optimizer

The optimization process for SABO is as follows.

Similar to other optimization algorithms, SABO randomly initializes the population within the search space. Mathematically, the population can be represented as a matrix, as shown in equation (9). The initial positions of the search agents in the search space are randomly generated using equation (10).

$$X = \begin{bmatrix} X_1 \\ \vdots \\ X_i \\ \vdots \\ X_N \end{bmatrix}_{N\times m} = \begin{bmatrix} X_{1,1} & \cdots & X_{1,d} & \cdots & X_{1,m} \\ \vdots & \ddots & \vdots & \iddots & \vdots \\ X_{i,1} & \cdots & X_{i,d} & \cdots & X_{i,m} \\ \vdots & \iddots & \vdots & \ddots & \vdots \\ X_{N,1} & \cdots & X_d & \cdots & X_{N,m} \end{bmatrix}_{N\times m} \tag{9}$$

$$x_{i,d} = lb_d + r_{i,d} \cdot (ub_d - lb_d)\,, i = 1, \ldots, N, d = 1, \ldots, m \tag{10}$$

Here, $X$ is the SABO population matrix, $X_i$ represents the i-th search agent, $x_{i,d}$ denotes its d-th dimension in the search space, N is the number of search agents, m is the number of decision variables $lb_d$ and $ub_d$ are the lower and upper bounds of the search space, respectively, and $r_{i,d}$ is a random number in the range of [0, 1].

Each search agent represents a candidate solution to the problem by proposing values for the decision variables. Thus, the objective function can be evaluated for each search agent. The evaluated values of the objective function are represented by a vector $\vec{F}$, as defined in equation (11). Based on the values assigned by each population member to the decision variables, the objective function is evaluated and stored in vector $\vec{F}$. Therefore, the number of elements in the vector $\vec{F}$ equals the number of population members N.

$$\vec{F} = \begin{bmatrix} F_1 \\ \vdots \\ F_i \\ \vdots \\ F_N \end{bmatrix}_{N\times 1} = \begin{bmatrix} F(x_1) \\ \vdots \\ F(x_i) \\ \vdots \\ F(x_N) \end{bmatrix}_{N\times 1} \tag{11}$$

Here, $\vec{F}$ is the vector of objective function values, and $F_i$ represents the evaluated objective function value for the i-th search agent.

The evaluated objective function values serve as a criterion for assessing the quality of the solutions proposed by the search agents. Thus, the best objective function value corresponds to the best search agent. Similarly, the worst objective function value corresponds to the worst search agent. Since the positions of the search agents are updated in each iteration, the best search agent is identified and saved until the algorithm's final iteration.

The SABO algorithm introduces a new computational concept, "$-_v$ ", referred to as the $v-$ subtraction of search agent B and search agent A, as defined in equation (12).

$$A -_v B = sign\left(F(A) - F(B)\right)\left(A - \vec{v} * B\right) \tag{12}$$

Here, $\vec{v}$ an m-dimensional vector whose components are random numbers from the set $\{1, 2\}$, the operation "$*$" denotes the Hadamard product, meaning that each element of the resulting vector is the product of the corresponding elements of the original vectors. $F(A)$ and $F(B)$ represent the objective function values of search agents A and B, respectively, and "sign" denotes the signum function. It is worth noting that, due to the use of a random vector $\vec{v}$ with components from the set $\{1, 2\}$, the result of the $v-$ subtraction operation is a point from a subset of the search space with a cardinality of $2^{m+1}$.

In SABO, the displacement of any search agent $X_i$ within the search space is calculated as the arithmetic mean of the differences between each search agent $X_j, j = 1, 2, \ldots, N$ and $v-$ subtracted from search agent $X_i$. The new position of each search agent is then calculated using equation (13).

$$X_i^{new} = X_i + \vec{r_i} * \frac{1}{N}\sum_{j=1}^{N}\left(X_i -_v X_j\right), i = 1, 2, \ldots, N \tag{13}$$

Here, $X_i^{new}$ represents the new proposed position for the i-th search agent $X_i$, N is the total number of the search agents, and $\vec{r_i}$ is an m-dimensional vector whose components follow a normal distribution within the interval [0, 1]. If the updated position yields a better objective function value, it replaces the original position; otherwise, the original position remains unchanged, as shown in equation (14).

$$X_i = \begin{cases} X_i^{new}, F_i^{new} < F_i \\ X_i, else \end{cases} \tag{14}$$

Here, $F_i$ and $F_i^{new}$ represent the objective function values of the search agents $X_i$ and $X_i^{new}$, respectively.

After updating all search agents, the first iteration of the algorithm is completed. Based on the newly evaluated positions and objective function values, the algorithm proceeds to the next iteration. In each iteration, the best search agent is saved as the best candidate solution found so far. This process continues until the algorithm reaches its final iteration. Finally, the best candidate solution saved during the iterations is presented as the solution to the problem.

### 2.3 Minimum envelope entropy

When using the SABO algorithm for optimization, it is essential to define a fitness function. Shannon entropy is an effective criterion for evaluating the sparsity characteristics of signals. The entropy value reflects the uniformity of the probability distribution, where the most uncertain distribution corresponds to the highest entropy [29]. Reference [30] introduces the concept of envelope entropy and transforms the envelope signal obtained after demodulation into a probability distribution sequence $p_j$. The entropy calculated from this sequence reflects the sparse characteristics of the original signal. The envelope entropy $E_p$ of zero-mean signal $x(j)\ (j = 1, 2, \cdots, N)$ is defined by equation (15).

$$\left.\begin{aligned} E_p &= -\sum_{j=1}^{N} p_j \lg p_j \\ p_j &= a(j) \Big/ \sum_{j=1}^{N} a(j) \end{aligned}\right\} \tag{15}$$

Here, $p_j$ is the normalized form of $a(j)$, and $a(j)$ represents the envelope signal obtained from $x(j)$ after Hilbert demodulation.

After decomposing the bearing fault signal, if an IMF contains significant noise and lacks clear periodic impact characteristics related to the fault, its sparsity decreases, and its envelope entropy increases. Conversely, if an IMF contains significant fault-related information and exhibits regular shock pulses in its waveform, it demonstrates strong sparsity and lower envelope entropy. To identify the globally optimal IMF that encapsulates the most significant fault-related information, we use the minimum envelope entropy as the fitness criterion in the optimization process. The ultimate goal is to minimize the envelope entropy.

### 2.4 The Kurtosis

After decomposing the fault signal using parameter-optimized FMD, we obtain n IMFs. The goal is to identify the IMF containing the most significant fault-related information.

Kurtosis is a dimensionless parameter used in time-domain analysis. Because kurtosis is independent of factors such as speed, size, and load of mechanical components and is highly sensitive to impact signals, it is highly effective in detecting surface damage in mechanical systems. When a device fails, the probability density of the signal increases due to shock vibrations, leading to a higher kurtosis value. A higher kurtosis value indicates that an IMF contains more fault-related components. Therefore, kurtosis is used as a criterion to select the final IMF for retention. This relationship is expressed in equation (16).

$$kurtosis = \frac{1}{N} \sum \left( \frac{x_i - \overline{x}}{\sigma} \right)^4 \tag{16}$$

Here, $x_i$ represents the fault signal, $\overline{x}$ is the mean value of the signal, and $\sigma$ is the standard deviation of the signal.

## 3 The proposed method and the comparison methods

### 3.1 The proposed method

This paper proposes a fault diagnosis method based on SABO and FMD. First, to address the issue that manual parameter selection in FMD reduces decomposition efficiency, the minimum envelope entropy is used as the fitness function for SABO to determine the optimal combination of filter length (L) and mode number (n) in FMD. Next, the optimal parameters are applied to FMD to decompose the signal into multiple IMFs. Then, the maximum kurtosis value is used to filter IMF

containing fault information. Finally, envelope spectrum analysis is applied to the selected IMF. The steps of the proposed method are as follows, and the flowchart is shown in Fig 1:

### 3.2 The comparison methods

After reading several references, two comparison methods are designed to compare with the method proposed in this paper.

Comparison method 1: First, the signal is decomposed using EMD. Next, the kurtosis value of each IMF is calculated. Finally, the IMF with the highest kurtosis value is retained, and its envelope spectrum is obtained.

Comparison method 2: First, based on reference [31], two key FMD parameters are set: the filter length (L = 40) and the number of modes (n = 2). The signal is then decomposed using FMD. Next, the kurtosis value of each IMF is calculated. Finally, the IMF with the highest kurtosis value is retained, and its envelope spectrum is obtained.

## 4 Data

The vibration data used in this study were sourced from the Case Western Reserve University Bearing Data Center. Fig 2 illustrates the experimental setup. The setup includes a 2 hp motor (left), a torque transducer/encoder (center), a dynamometer (right), and control electronics (not shown). Single-point faults were introduced into the test bearings using electro-discharge machining, with fault diameters of 7, 14, 21, 28, and 40 mils (1 mil = 0.001 inches). Two bearings were

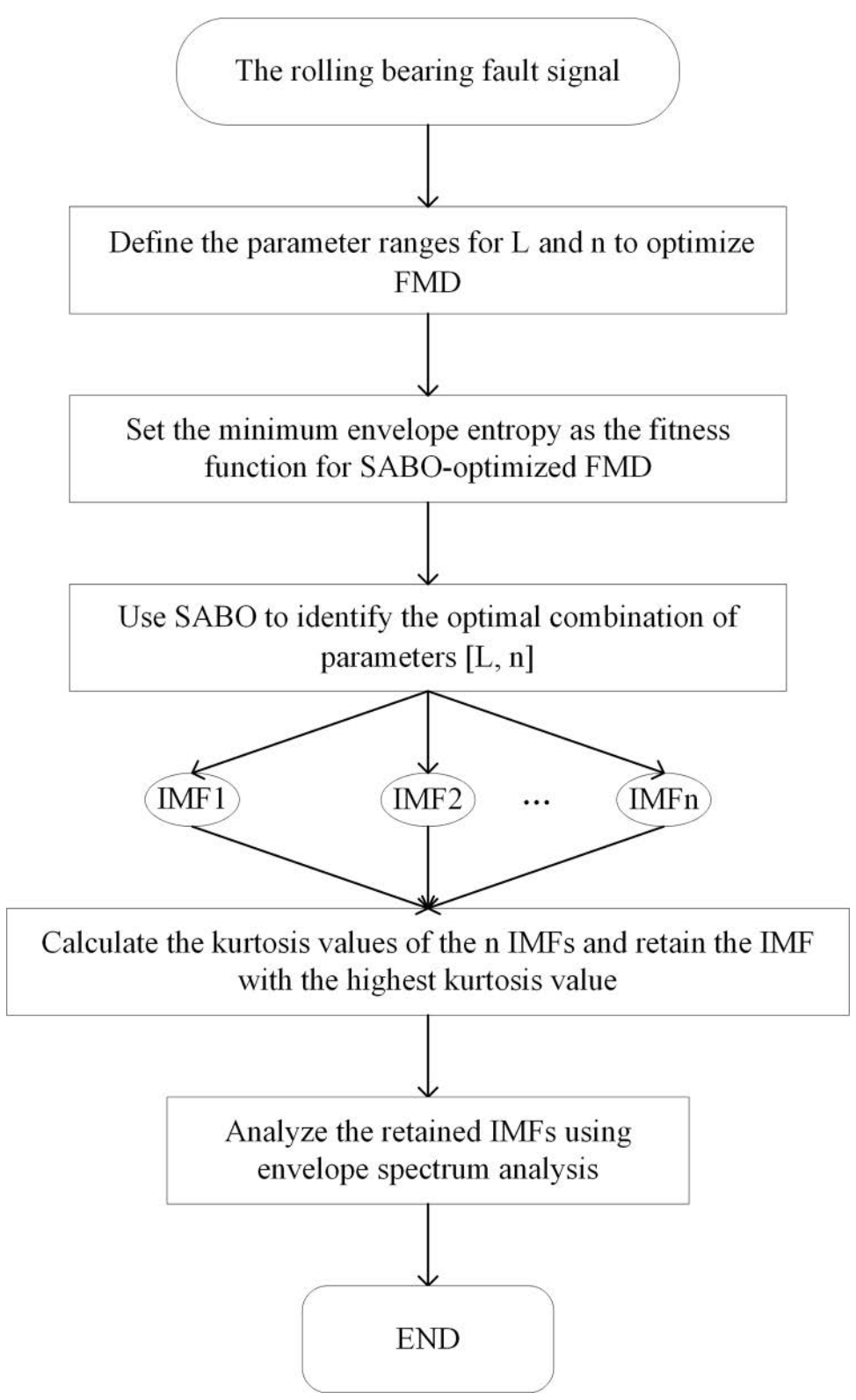


**Fig 1. The flowchart of the proposed method.** Step 1: Load the raw signal and define the parameter ranges for L and n to optimize FMD. Based on reference [22], the range for L is [2, 50], and the range for n is [2, 7]. Step 2: Set the minimum envelope entropy as the fitness function for SABO-optimized FMD. Step 3: Use SABO to identify the optimal combination of parameters [L, n]. Step 4: Calculate the kurtosis values of the n IMFs and retain the IMF with the highest kurtosis value. Step 5: Analyze the retained IMFs using envelope spectrum analysis.


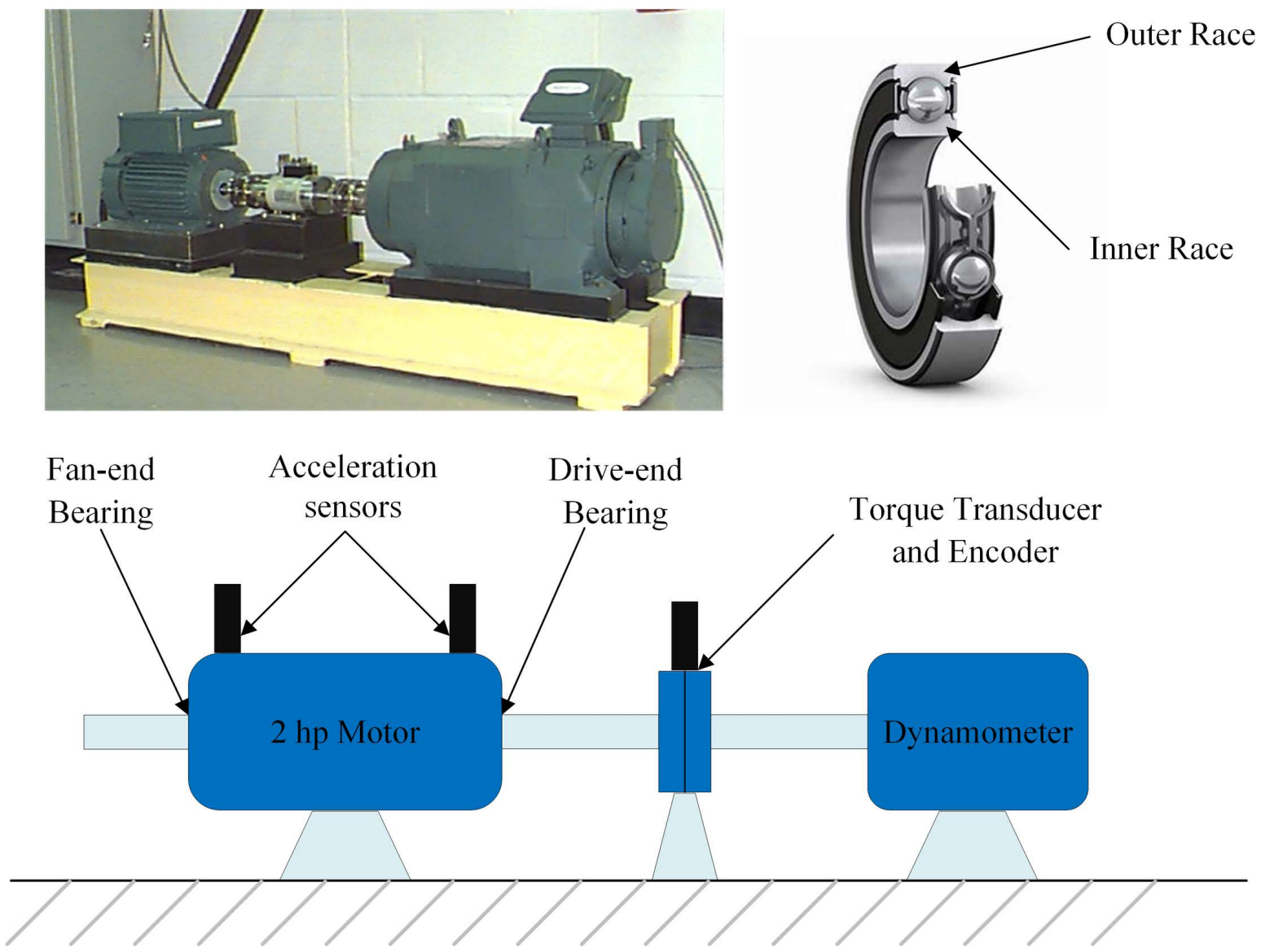


**Fig 2. The experimental apparatus used to extract the experimental data and its schematics.**


tested: the drive-end bearing and the fan-end bearing. The experimental data used in this study are from the drive-end bearing. The key parameters of the drive-end bearing are listed in Table 1.

When a bearing fails, vibration pulses generate specific frequencies. Different fault types exhibit distinct characteristic frequencies. The characteristic frequencies for different fault types are provided in Table 2.

## 5 Results and discussion

### 5.1 Experiment one

Experiment One used the dataset 170.mat, as described in Table 3.

**5.1.1 Result of experiment one.** The iterative process of SABO is illustrated in Fig 3. As shown in Fig 3, after two iterations, the objective function value stabilizes at its minimum, corresponding to the optimal parameters L = 130 and n = 2.

The kurtosis values of the two IMFs are computed, and the results are $kurtosis_{IMF1} = 3.79$ and $kurtosis_{IMF2} = 42.85$.The envelope spectrum of IMF2, which has a high kurtosis value, is shown in Fig 4 and its time-domain waveform is displayed in Fig 5. As shown in Fig 4, the horizontal coordinate of the highest peak is 160.038 Hz, which is close to the theoretical inner ring fault frequency of the bearing. This indicates an inner ring fault in the bearing. The horizontal coordinate of the

**Table 1. The basic parameters of the drive-end bearing.**

| Inside Diameter | Outside Diameter | Thickness | Ball Diameter | Pitch Diameter | Frequency |
|---|---|---|---|---|---|
| 0.9843 inches | 2.0472 inches | 0.5906 inches | 0.3126 inches | 1.537 inches | 12000 Hz |


**Table 2. The fault characteristic frequencies (multiple of running speed in Hz).**

| Inner Ring | Outer Ring |
|---|---|
| 5.4152 | 3.5848 |


Table 3. The data description of 170.mat.

| Fault type | Fault Diameter | Motor Load | Motor Approximate Speed | Motor Rotation Frequency | Failure Frequency |
|---|---|---|---|---|---|
| Inner ring failure | 0.3556mm | 1 hp | 1772 r/min | 29.533 Hz | 159.929 Hz |


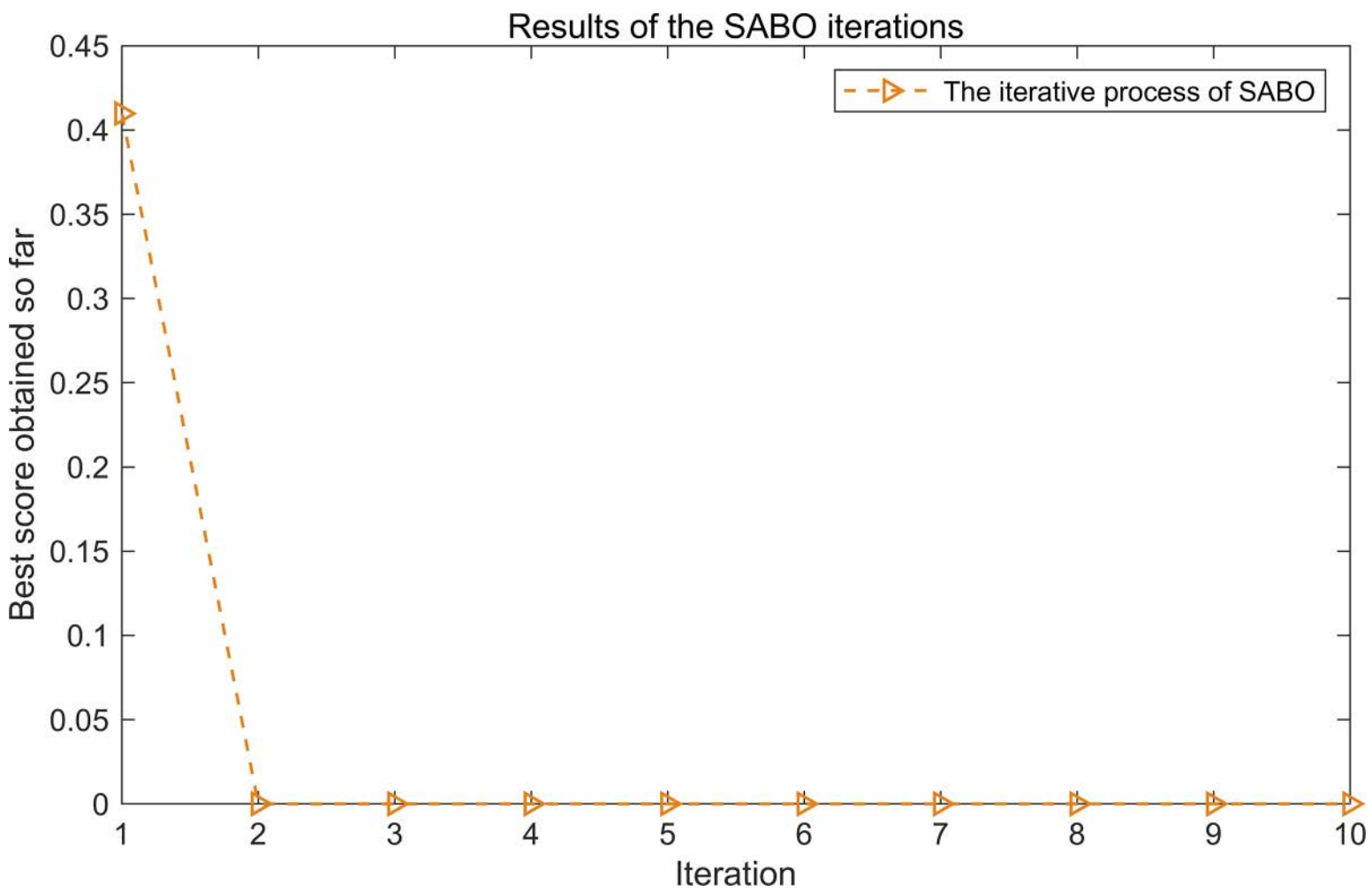


Fig 3. The iterative process of the SABO.


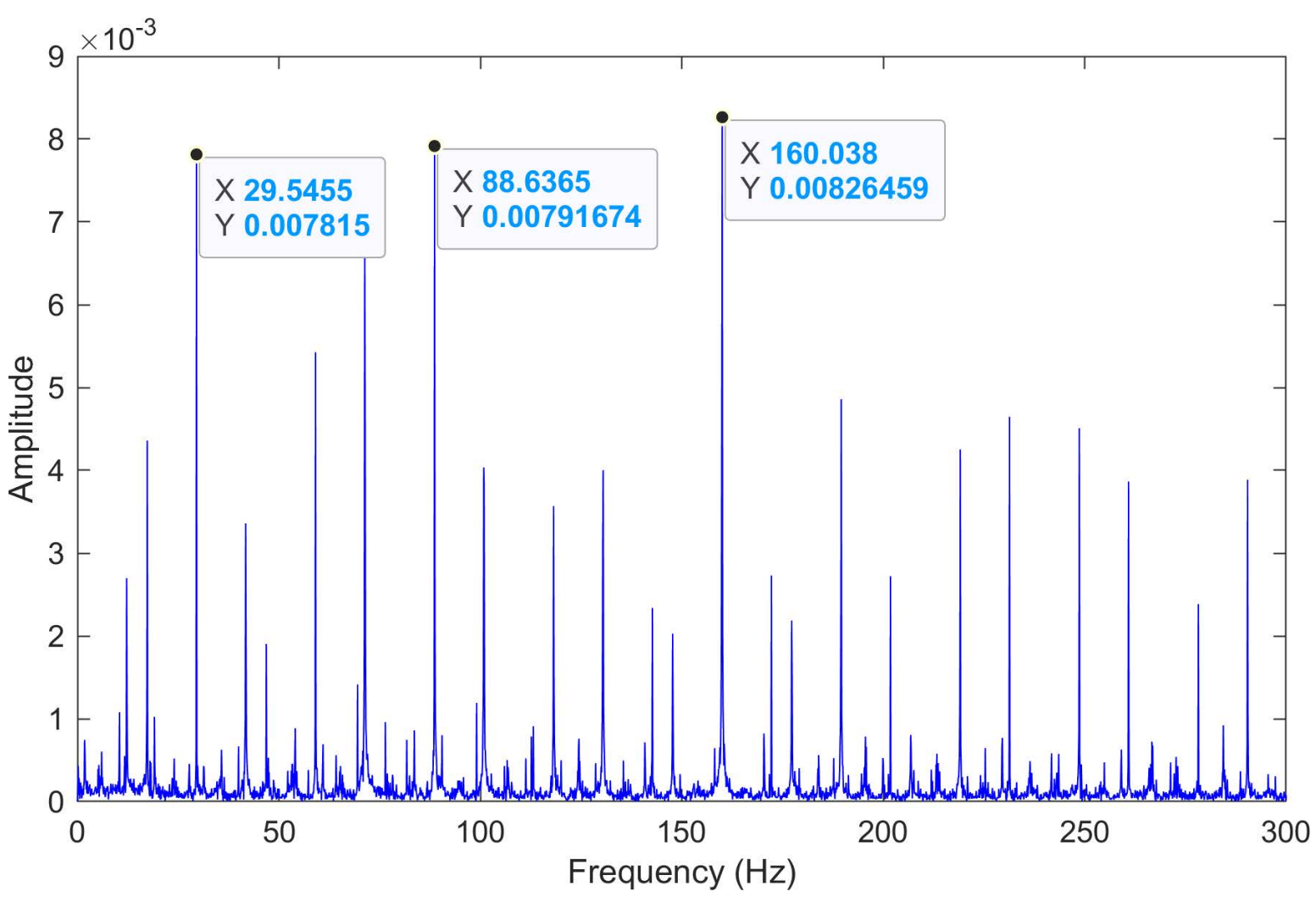


Fig 4. The envelope spectrum after processing by the proposed method of 170.mat.


second-highest peak is 88.6365 Hz, which is close to the third harmonic (3×) of the bearing's rotational frequency. The horizontal coordinate of the third-highest peak is 29.5455 Hz, which matches the bearing's rotational frequency.

The envelope spectrum of the comparison method 1 is shown in Fig 6 and its time-domain waveform is displayed in Fig 7. As shown in Fig 6, the horizontal coordinate of the highest peak is 29.2969 Hz, which matches the bearing's

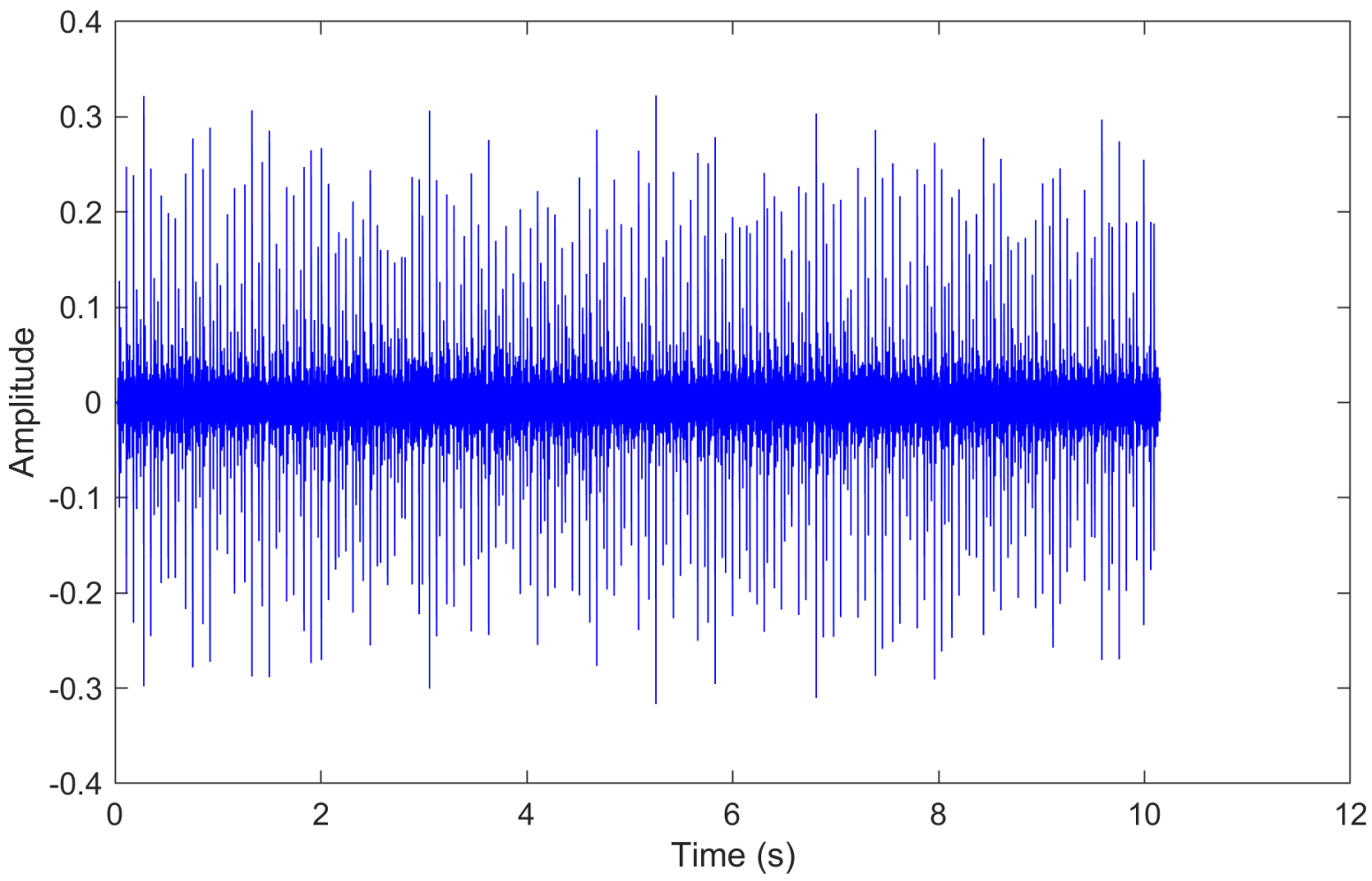


**Fig 5. The time-domain waveform after processing by the proposed method of 170.mat.**


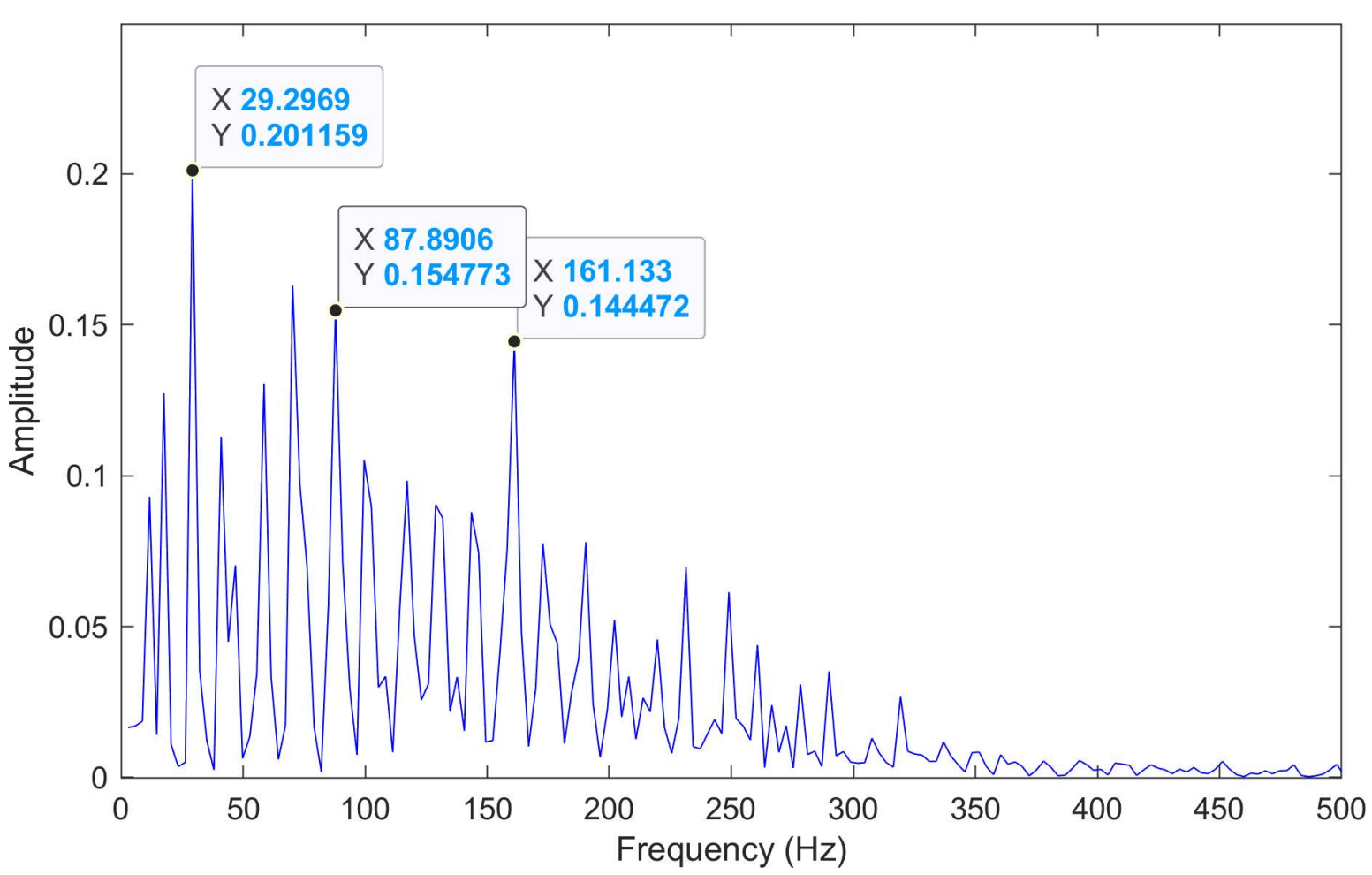


**Fig 6. The envelope spectrum after processing by the comparison method 1 of 170.mat.**


rotational frequency. The horizontal coordinate of the second-highest peak is 87.8906 Hz, which is close to the third harmonic (3×) of the bearing's rotational frequency. The horizontal coordinate of the third-highest peak is 161.133 Hz, which is close to the inner ring fault frequency of the bearing. Compared to the other two labeled peaks, the peak corresponding to the bearing's inner ring fault frequency is less prominent, making the fault information easy to overlook.

The envelope spectrum of the comparison method 2 is shown in Fig 8 and its time-domain waveform is displayed in Fig 9. As shown in Fig 8, the horizontal coordinate of the highest peak is 29.5455 Hz, which matches the bearing's rotational frequency. The horizontal coordinate of the other labeled peak is 160.038 Hz, which is close to the inner ring fault frequency of the bearing. Compared to the highest peaks, the peaks near the bearing's inner ring fault frequency are less prominent, making the fault information easy to overlook.

**5.1.2 Discussion of experiment one.** The experimental results for the three methods are summarized in Table 4. By comparing the results of the three methods, two conclusions can be drawn. (1) Compared with comparison method

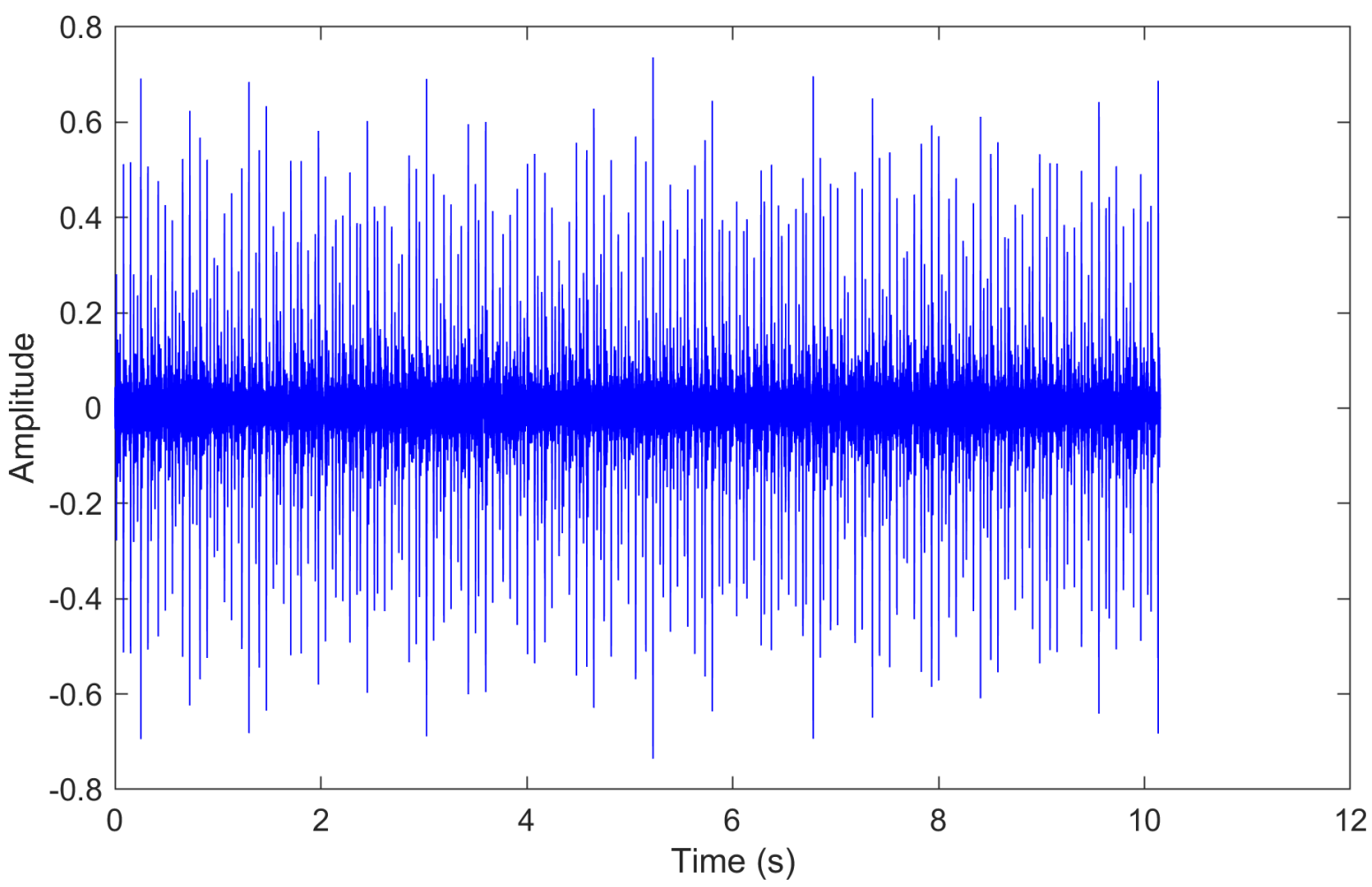


**Fig 7. The time-domain waveform after processing by the comparison method 1 of 170.mat.**


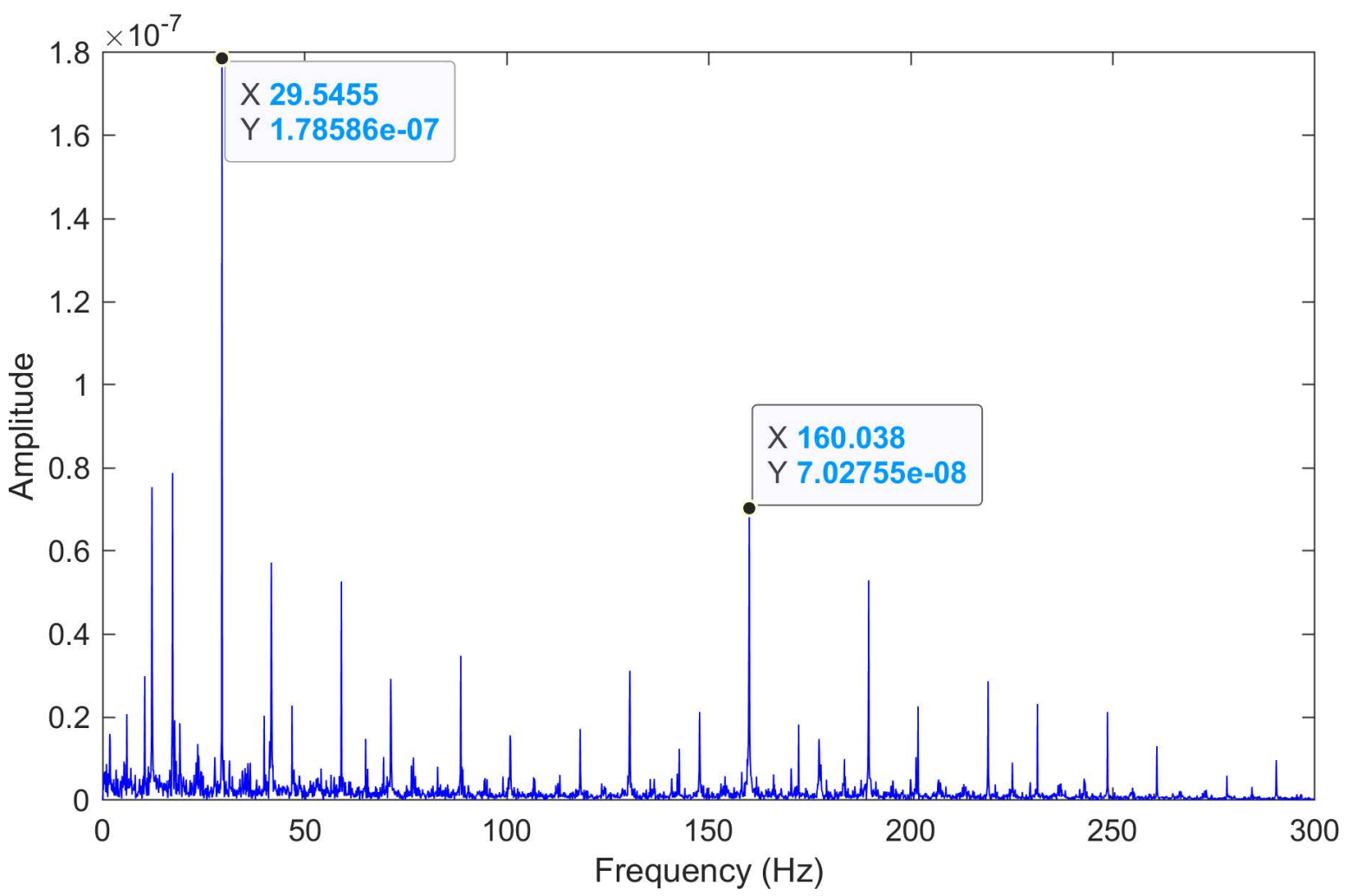


**Fig 8. The envelope spectrum after processing by the comparison method 2 of 170.mat.**


1, the proposed method achieves a lower error rate in fault frequency identification. (2) Compared to the two benchmark methods, the proposed method extracts fault information with the highest clarity.

### 5.2 Experiment two

Experiment Two used the dataset 210.mat, as described in Table 5.

**5.2.1 Result of experiment two.** The iterative process of SABO is illustrated in Fig 10. As shown in Fig 10, after two iterations, the objective function value stabilizes at its minimum, corresponding to the optimal parameters L = 80 and n = 3.

The kurtosis values of the three IMFs are computed, and the results are $kurtosis_{IMF1} = 13.1$,$kurtosis_{IMF2} = 13.6$ and $kurtosis_{IMF3} = 15.6$.The envelope spectrum of IMF3, which has a high kurtosis value, is shown in Fig 11 and its time-domain waveform is displayed in Fig 12. As shown in Fig 11, the horizontal coordinate of the highest peak is 159.828 Hz,

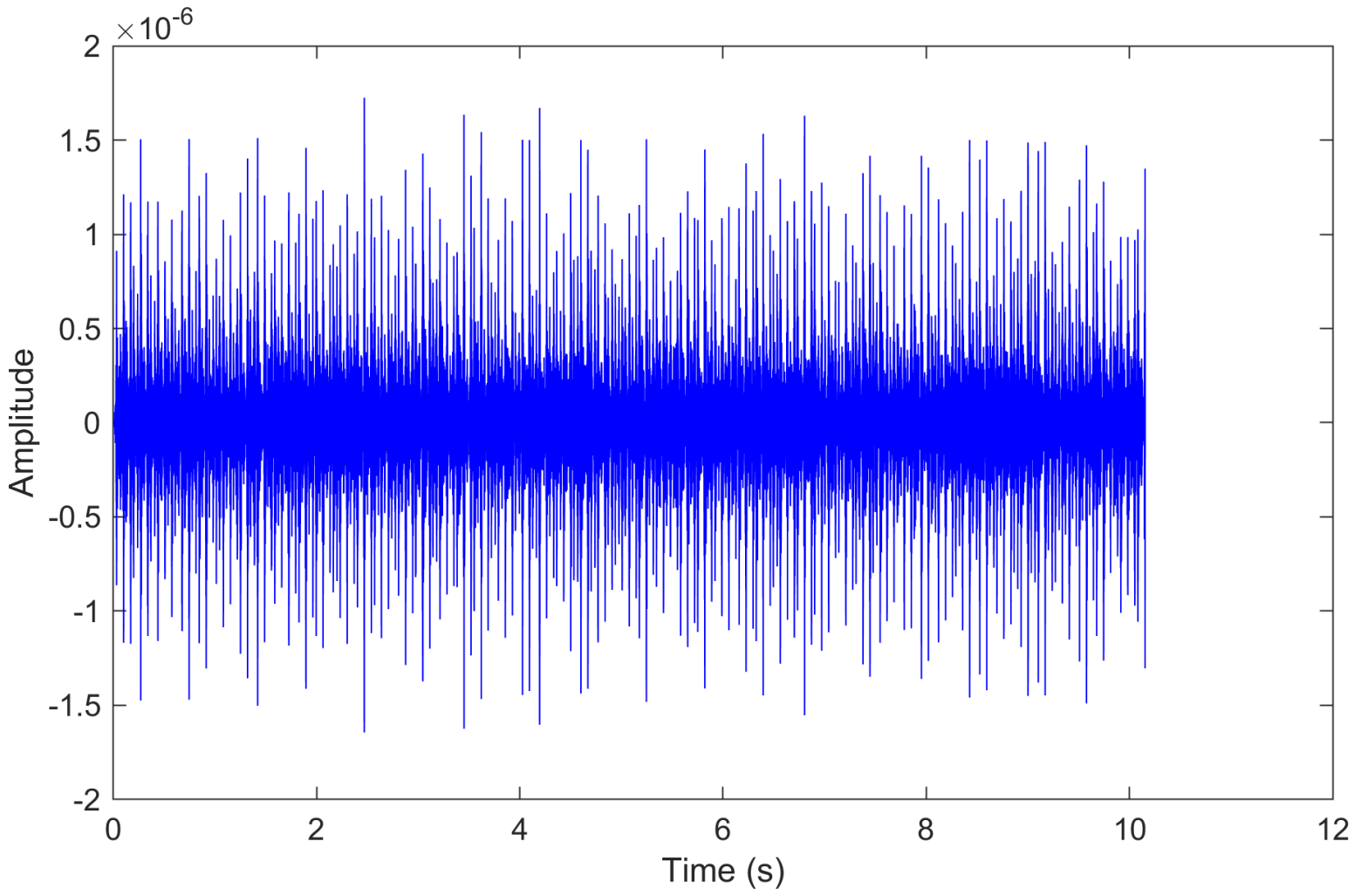


**Fig 9. The time-domain waveform after processing by the comparison method 2 of 170.mat.**


**Table 4. The experimental results of 170.mat.**

| / | Theoretical value(Hz) | The proposed method(Hz) | Error rate(%) | Comparison method 1(Hz) | Error rate(%) | Comparison method 2(Hz) | Error rate(%) |
|---|---|---|---|---|---|---|---|
| $f_R$ | 29.533 | 29.545 | 0.041 | 29.297 | 0.081 | 29.545 | 0.041 |
| $3f_R$ | 88.6 | 88.636 | 0.041 | 87.891 | 0.807 | / | / |
| $f_I$ | 159.929 | 160.038 | 0.069 | 161.133 | 0.748 | 160.038 | 0.069 |


**Table 5. The data description of 210.mat.**

| Fault type | Fault Diameter | Motor Load | Motor Approx-imate Speed | Motor Rotation Frequency | Failure Frequency |
|---|---|---|---|---|---|
| Inner ring failure | 0.3556mm | 1 hp | 1772 r/min | 29.533 Hz | 159.929 Hz |


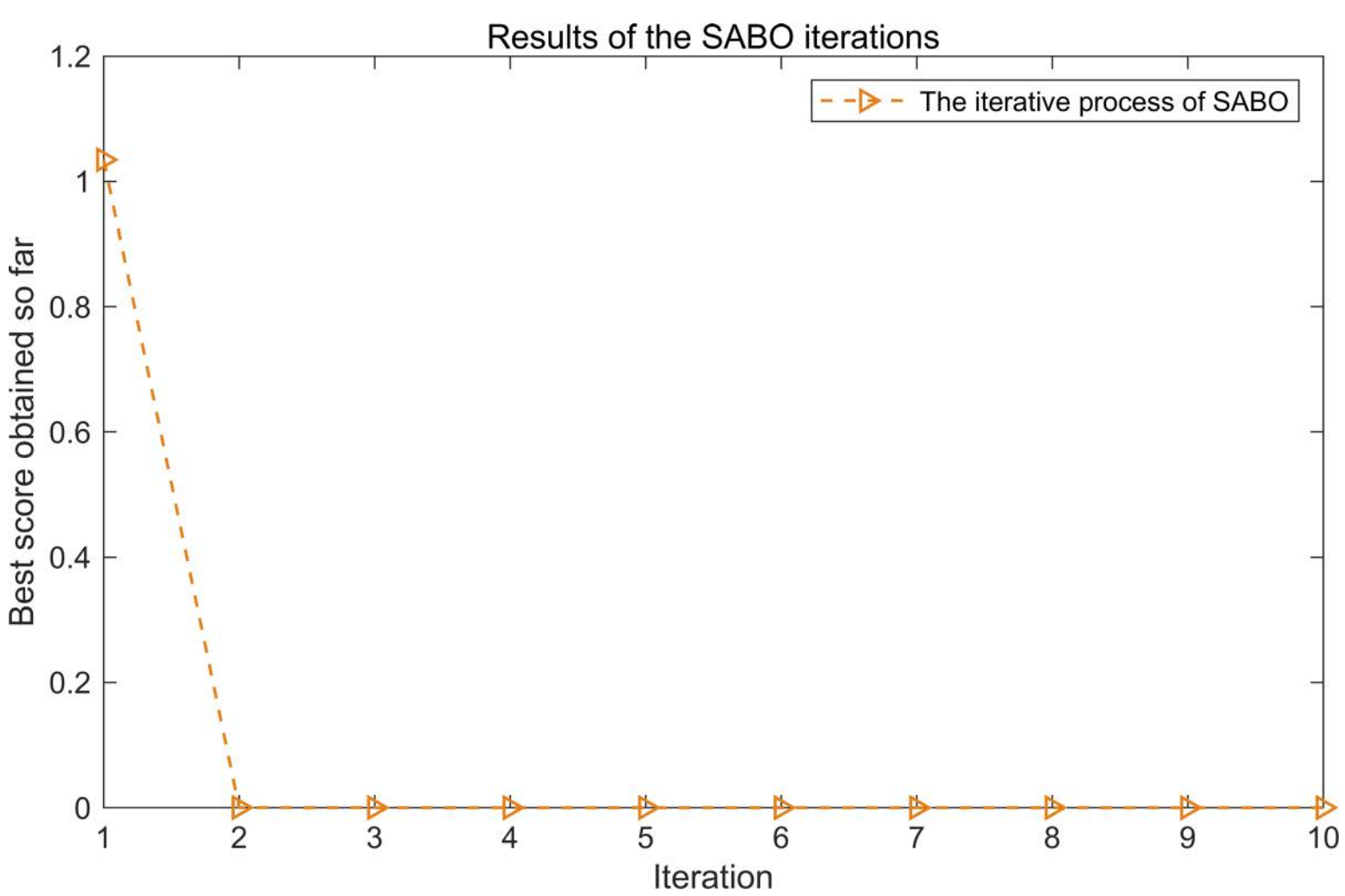


**Fig 10. The iterative process of the SABO.**


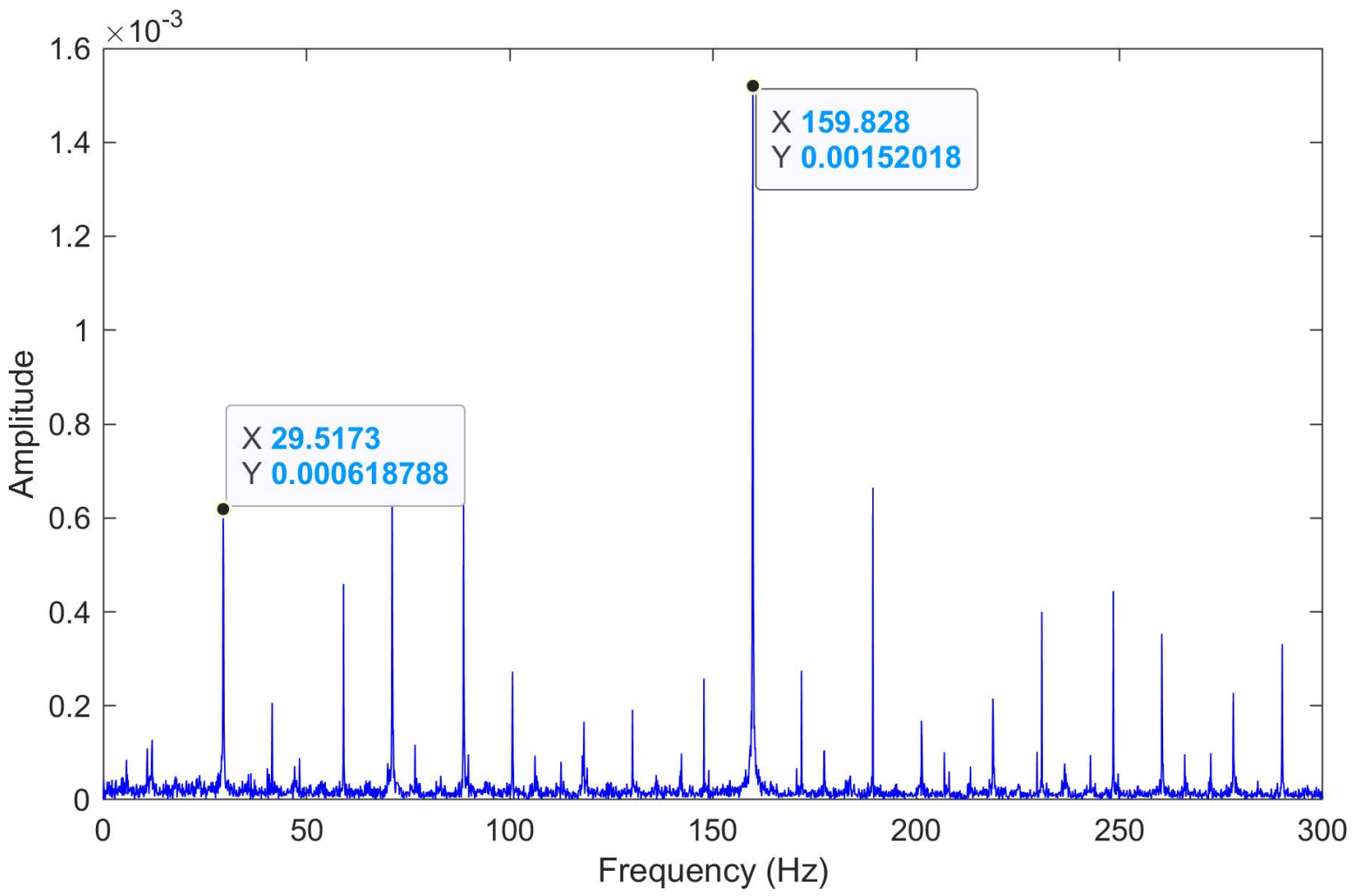


**Fig 11. The envelope spectrum after processing by the proposed method of 210.mat.**


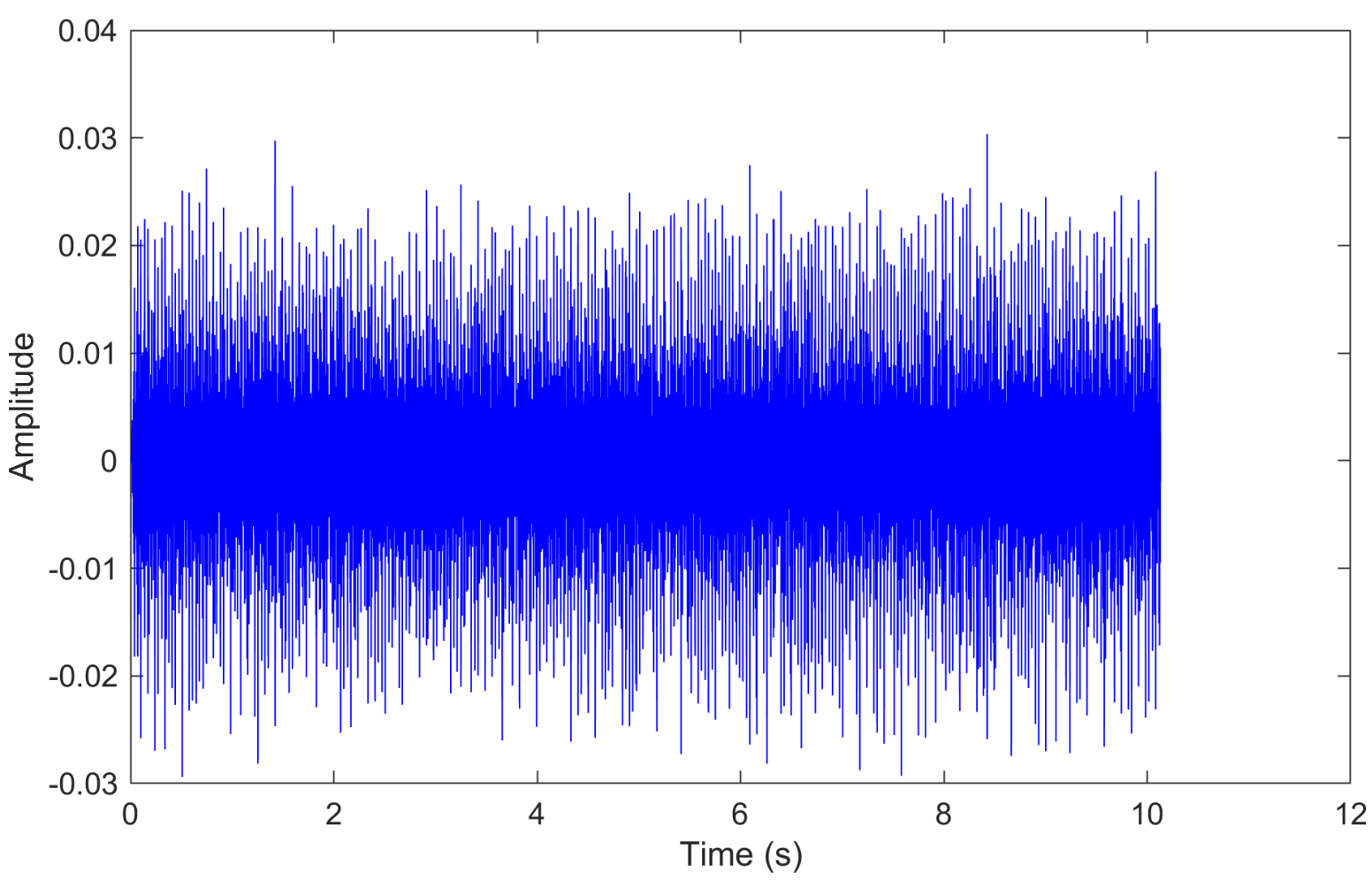


**Fig 12. The time-domain waveform after processing by the proposed method of 210.mat.**


which is close to the theoretical inner ring fault frequency of the bearing. This indicates an inner ring fault in the bearing. The horizontal coordinate of the other labeled peak is 29.5173 Hz, which matches the bearing's rotational frequency.

The envelope spectrum of the comparison method 1 is shown in Fig 13 and its time-domain waveform is displayed in Fig 14. As shown in Fig 13, the horizontal coordinate of the highest peak is 161.133 Hz, which is close to the theoretical inner ring fault frequency of the bearing. The horizontal coordinate of the second-highest peak is 29.2969 Hz, which matches the bearing's rotational frequency. The horizontal coordinate of the third-highest peak is 319.336 Hz, which is close to the second harmonic (2×) of the inner ring fault frequency.

The envelope spectrum of the comparison method 2 is shown in Fig 15 and its time-domain waveform is displayed in Fig 16. As shown in Fig 15, the horizontal coordinate of the highest peak is 29.5173 Hz, which matches the bearing's

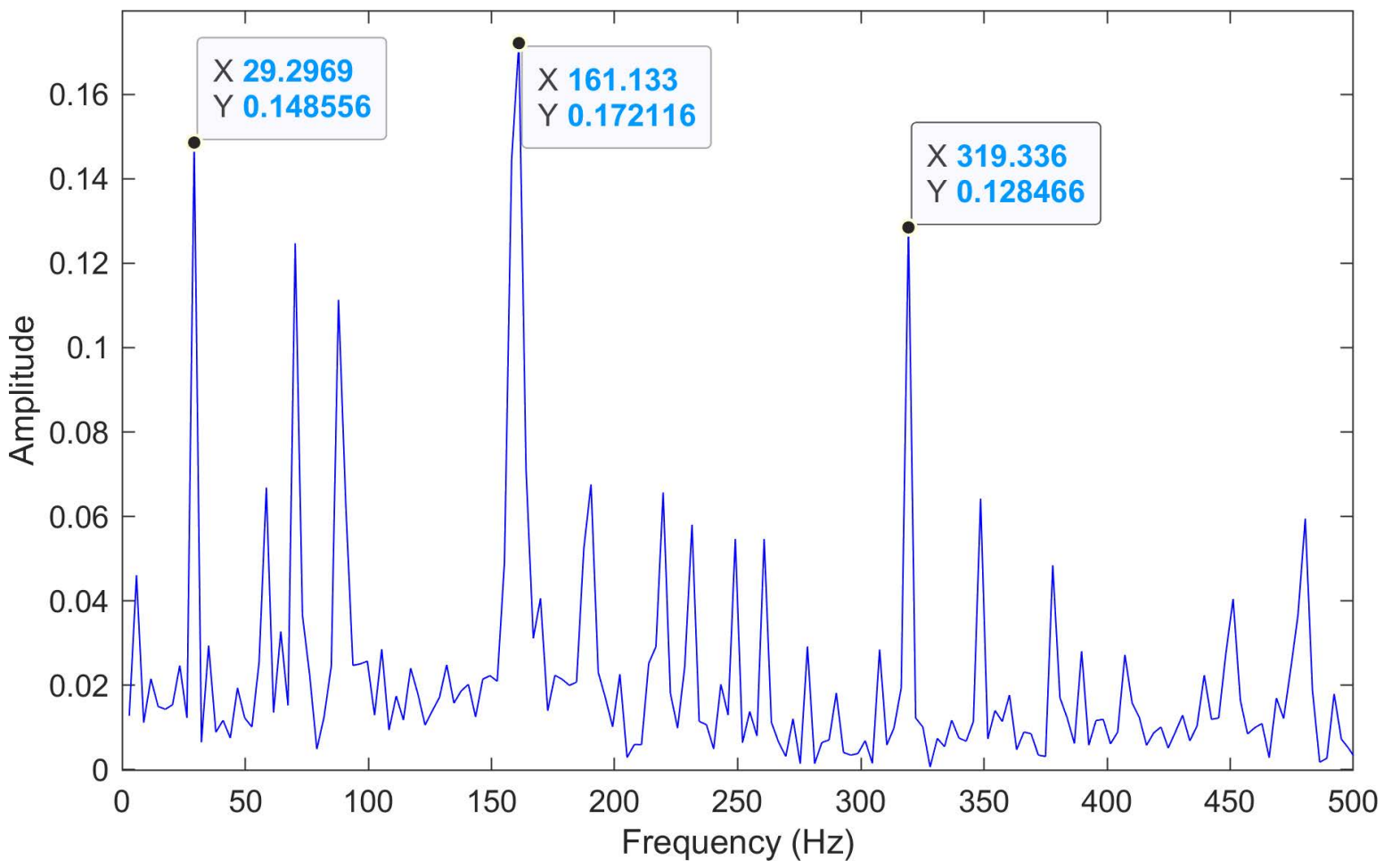


**Fig 13. The envelope spectrum after processing by the comparison method 1 of 210.mat.**


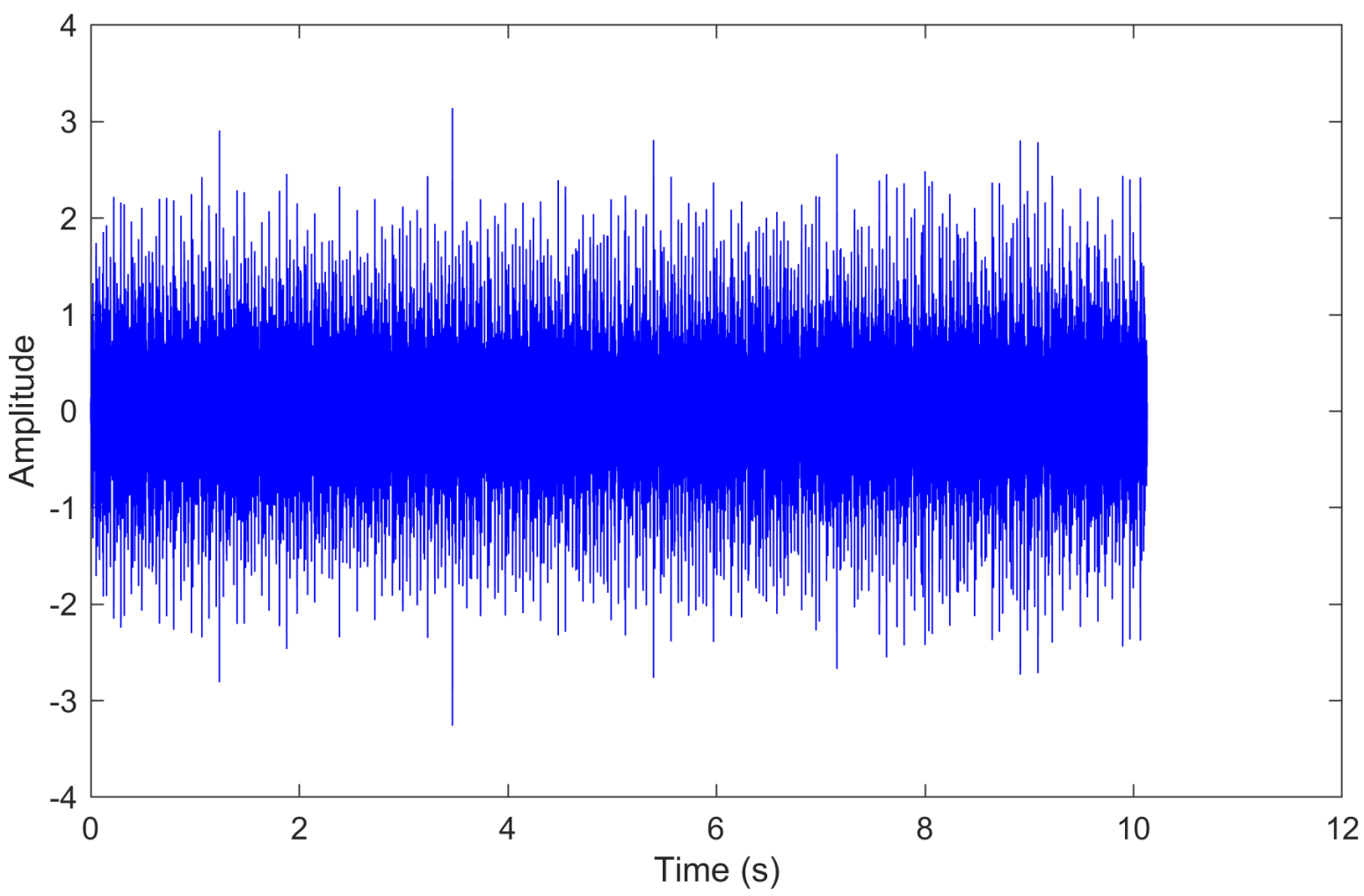


**Fig 14. The time-domain waveform after processing by the comparison method 1 of 210.mat.**


rotational frequency. The horizontal coordinate of the other labeled peak is 159.828 Hz, which is close to the inner ring fault frequency of the bearing. Compared to the highest peaks, the peaks near the bearing's inner ring fault frequency are less prominent, making the fault information easy to overlook.

**5.2.2 Discussion of experiment two.** The experimental results for the three methods are summarized in Table 6. B By comparing the results of the three methods, two conclusions can be drawn. (1) Compared with the comparison method 1, the proposed method achieves a lower error rate in fault frequency identification. (2) Compared to the two benchmark methods, the proposed method extracts fault information with the highest clarity.

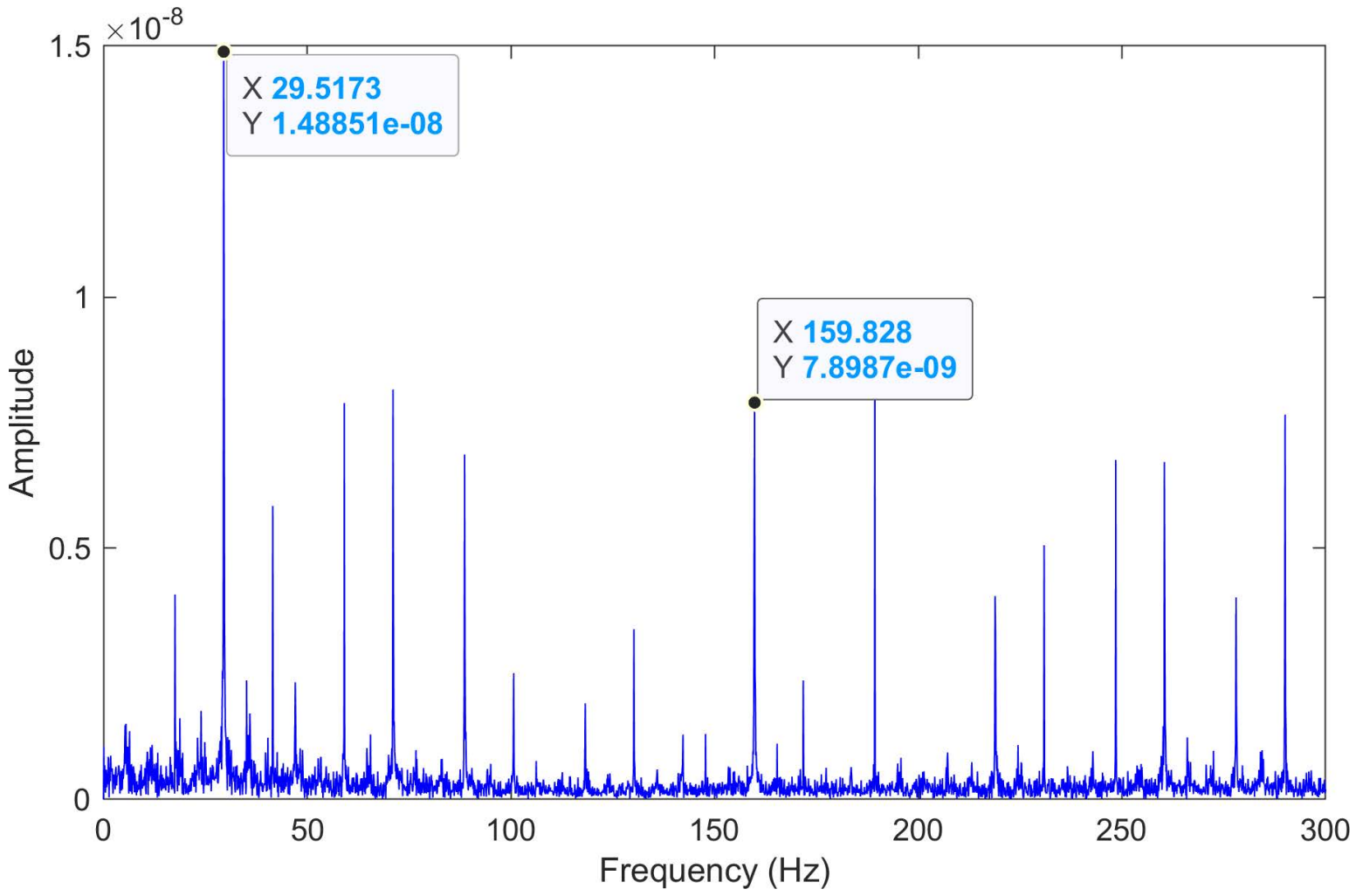


**Fig 15. The envelope spectrum after processing by the comparison method 2 of 210.mat.**


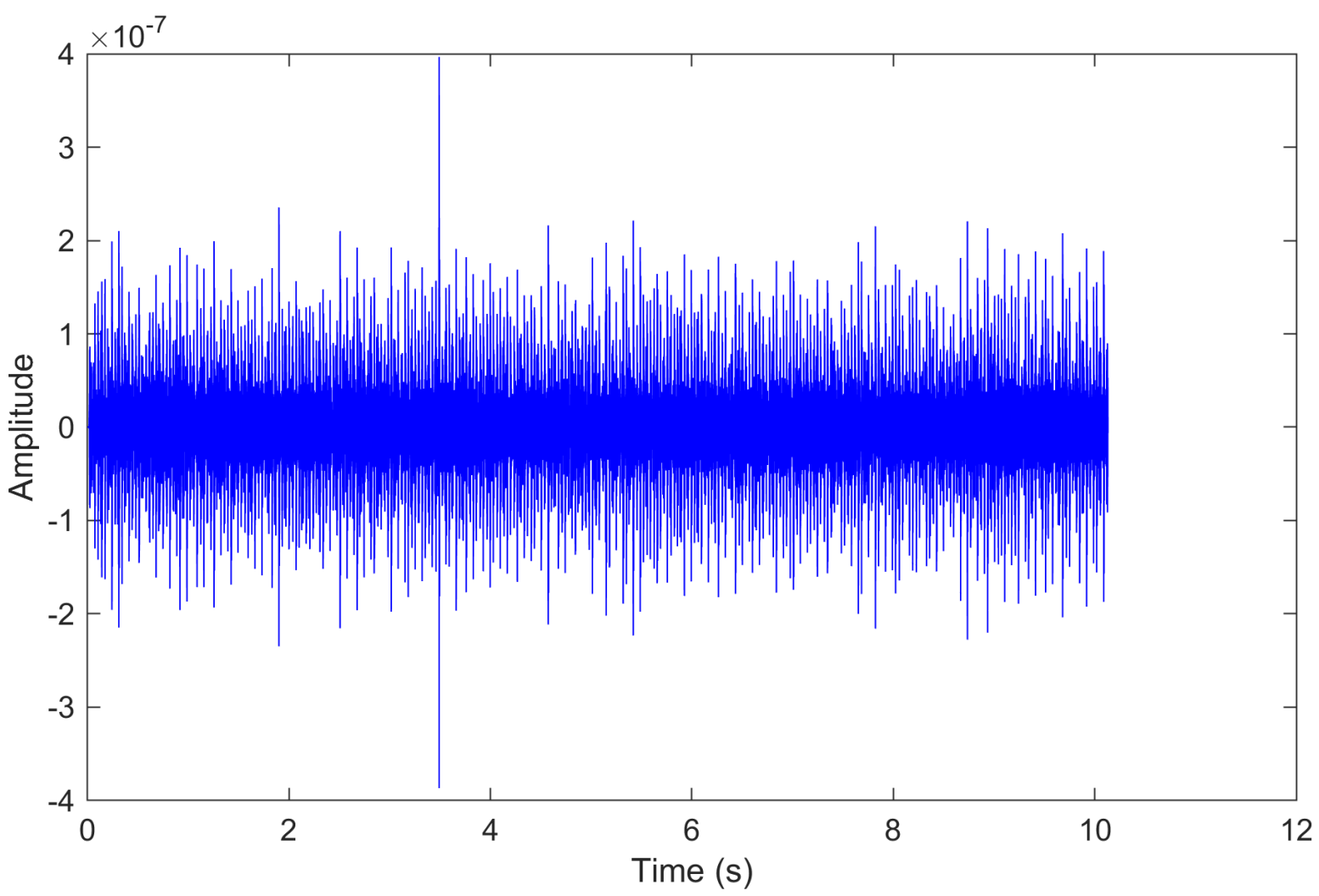


**Fig 16. The time-domain waveform after processing by the comparison method 2 of 210.mat.**


**Table 6. The experimental results of 210.mat.**

| / | Theoretical value(Hz) | The proposed method(Hz) | Error rate(%) | Comparison method 1(Hz) | Error rate(%) | Comparison method 2(Hz) | Error rate(%) |
|---|---|---|---|---|---|---|---|
| $f_R$ | 29.533 | 29.517 | 0.054 | 29.297 | 0.081 | 29.517 | 0.054 |
| $f_I$ | 159.929 | 159.828 | 0.063 | 161.133 | 0.748 | 159.828 | 0.063 |
| $2f_I$ | 319.858 | / | / | 319.336 | 0.162 | / | / |


## 5.3 Experiment three

Experiment Three used the dataset 130.mat, as described in Table 7.

**5.3.1 Result of experiment three.** The iterative process of SABO is illustrated in Fig 17. As shown in Fig 17, after three iterations, the objective function value stabilizes at its minimum, corresponding to the optimal parameters L = 30 and n = 3.

The kurtosis values of the three IMFs are computed, and the results are $kurtosis_{IMF1} = 2.9$,$kurtosis_{IMF2} = 14.9$ and $kurtosis_{IMF3} = 15.2$.The envelope spectrum of IMF3, which has a high kurtosis value, is shown in Fig 18 and its time-domain waveform is displayed in Fig 19. As shown in Fig 18, the horizontal coordinate of the highest peak is 107.614 Hz, which is close to the theoretical outer ring fault frequency of the bearing. This indicates an outer ring fault in the bearing. The horizontal coordinate of the second-highest peak is 215.327 Hz, which is close to the second harmonic (2×) of the outer ring fault frequency. The horizontal coordinate of the third-highest peak is 29.9038 Hz, which matches the bearing's rotational frequency.

The envelope spectrum of the comparison method 1 is shown in Fig 20 and its time-domain waveform is displayed in Fig 21. As shown in Fig 20, the horizontal coordinate of the highest peak is 108.398 Hz, which is close to the theoretical outer ring fault frequency of the bearing. This indicates an outer ring fault in the bearing. The horizontal coordinate of the second-highest peak is 216.797 Hz, which is close to the second harmonic (2×) of the outer ring fault frequency. The horizontal coordinate of the third-highest peak is 29.2969 Hz, which matches the bearing's rotational frequency.

The envelope spectrum of the comparison method 2 is shown in Fig 22 and its time-domain waveform is displayed in Fig 23. As shown in Fig 22, the horizontal coordinate of the highest peak is 29.5173 Hz, which matches the bearing's

**Table 7. The data description of 130.mat.**

| Fault type | Fault Diameter | Motor Load | Motor Approx-imate Speed | Motor Rotation Frequency | Failure Frequency |
|---|---|---|---|---|---|
| Outer ring failure | 0.1778mm | 0 hp | 1797 r/min | 29.95 Hz | 107.365 Hz |


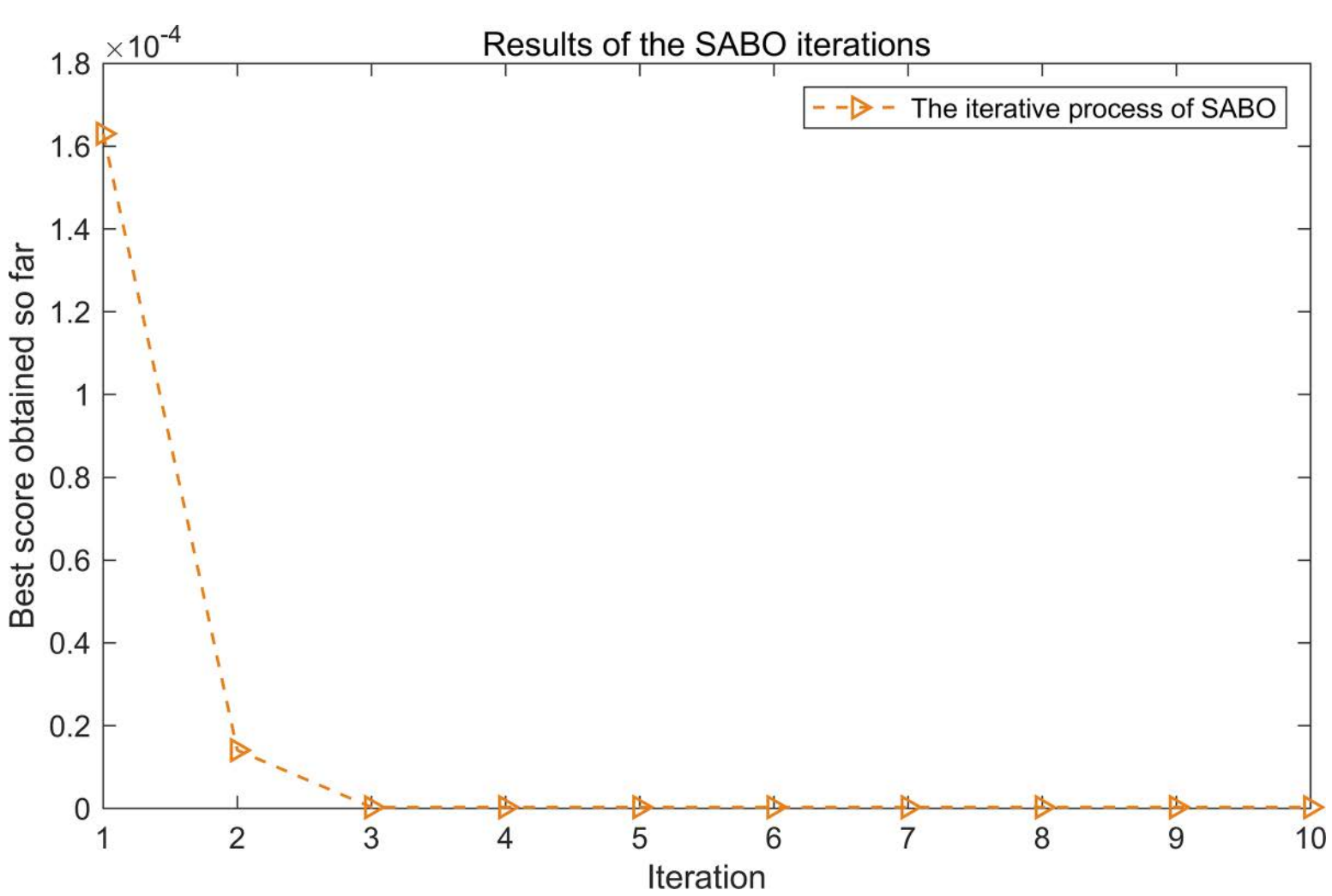


**Fig 17. The iterative process of the SABO.**


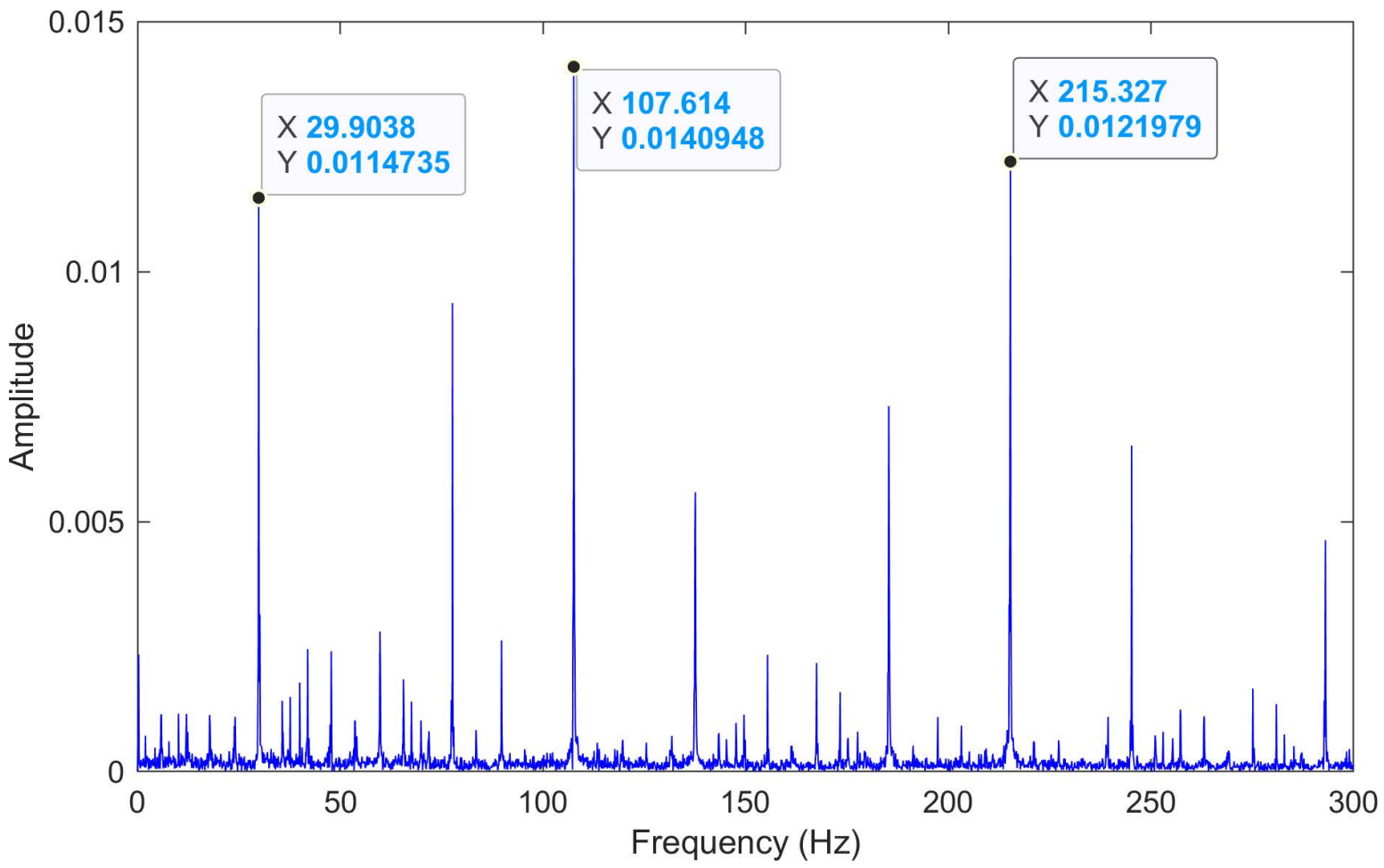


**Fig 18. The envelope spectrum after processing by the proposed method of 130.mat.**


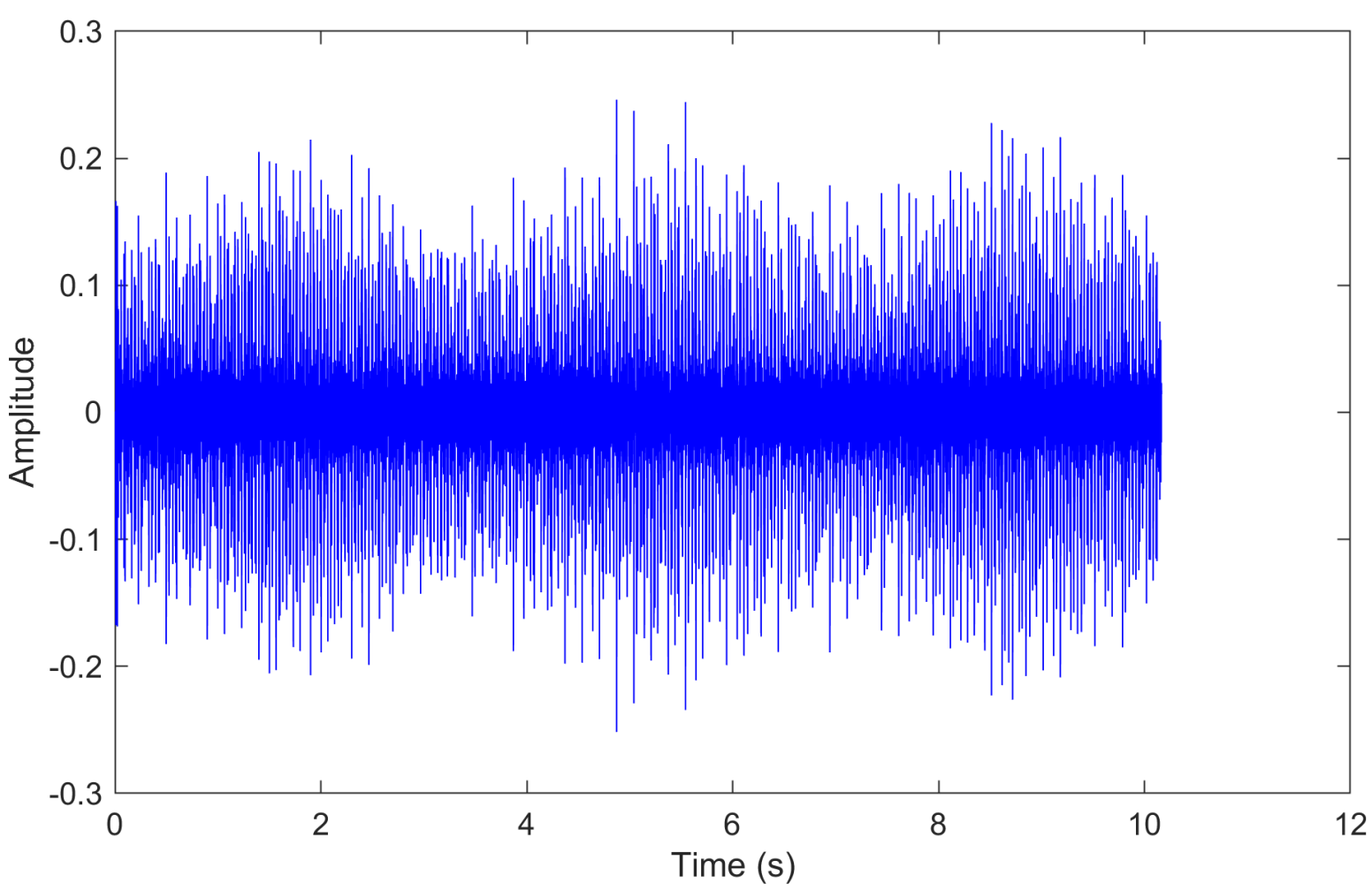


**Fig 19. The time-domain waveform after processing by the proposed method of 130.mat.**


rotational frequency. The horizontal coordinate of the second-highest peak is 215.327 Hz, which is close to the second harmonic (2×) of the outer ring fault frequency. The horizontal coordinate of the other labeled peak is 107.614 Hz, which is close to the outer ring fault frequency of the bearing.

**5.3.2 Discussion of experiment three.** The experimental results for the three methods are summarized in Table 8. By comparing the results of the three methods, two conclusions can be drawn. (1) Compared with the comparison method 1, the proposed method achieves a lower error rate in fault frequency identification. (2) Although both the proposed method and the comparison method 2 identify the same fault frequencies, the fault information obtained by the proposed method is more pronounced.

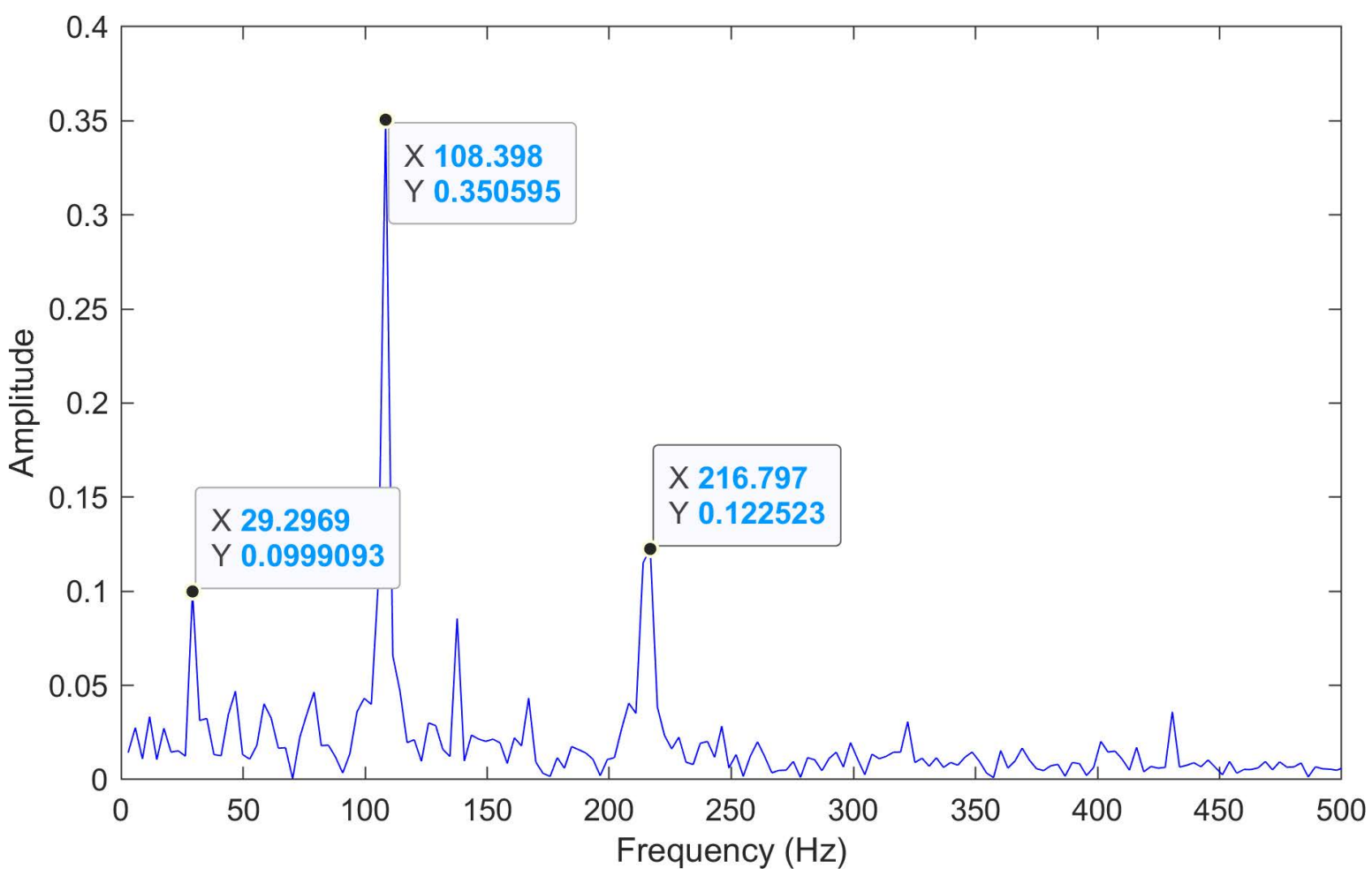


**Fig 20. The envelope spectrum after processing by the comparison method 1 of 130.mat.**


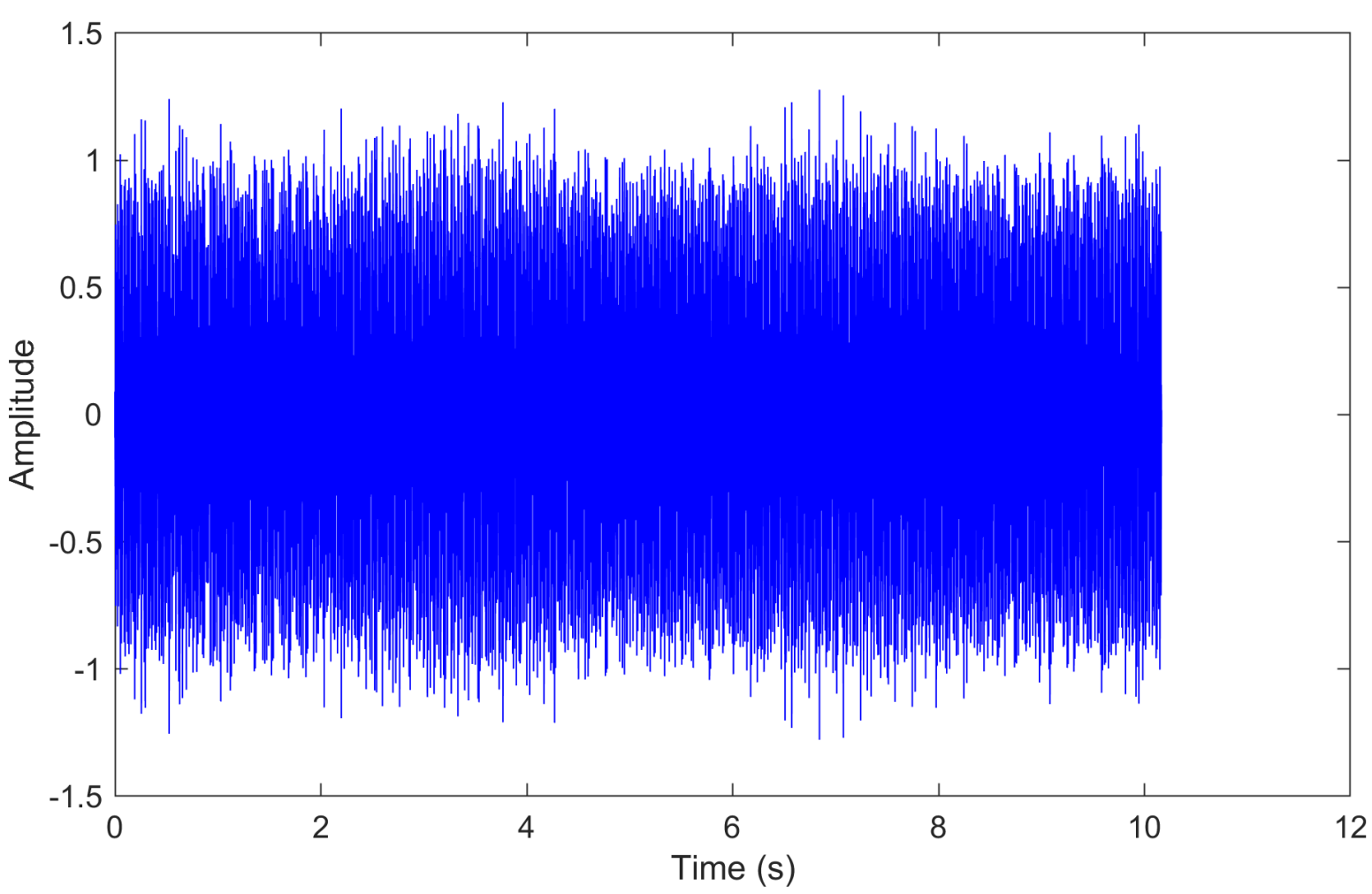


**Fig 21. The time-domain waveform after processing by the comparison method 1 of 130.mat.**


## 5.4 Experiment four

**5.4.1 Result of experiment four.** Experiment Four used the dataset 236.mat, as described in Table 9.

The iterative process of SABO is illustrated in Fig 24. after eight iterations, the objective function value stabilizes at its minimum, corresponding to the optimal parameters L = 30 and n = 2.

The kurtosis values of the two IMFs are computed, and the results are $kurtosis_{IMF1} = 44.9$ and $kurtosis_{IMF2} = 451.9$ .The envelope spectrum of IMF3, which has a high kurtosis value, is shown in Fig 25 and its time-domain waveform is displayed in Fig 26. As shown in Fig 25, the horizontal coordinate of the highest peak is 104.612 Hz, which is close to the theoretical outer ring fault frequency of the bearing. This indicates an outer ring fault in the bearing. The horizontal coordinate of the second-highest peak is 29.146 Hz, which matches the bearing's rotational frequency.

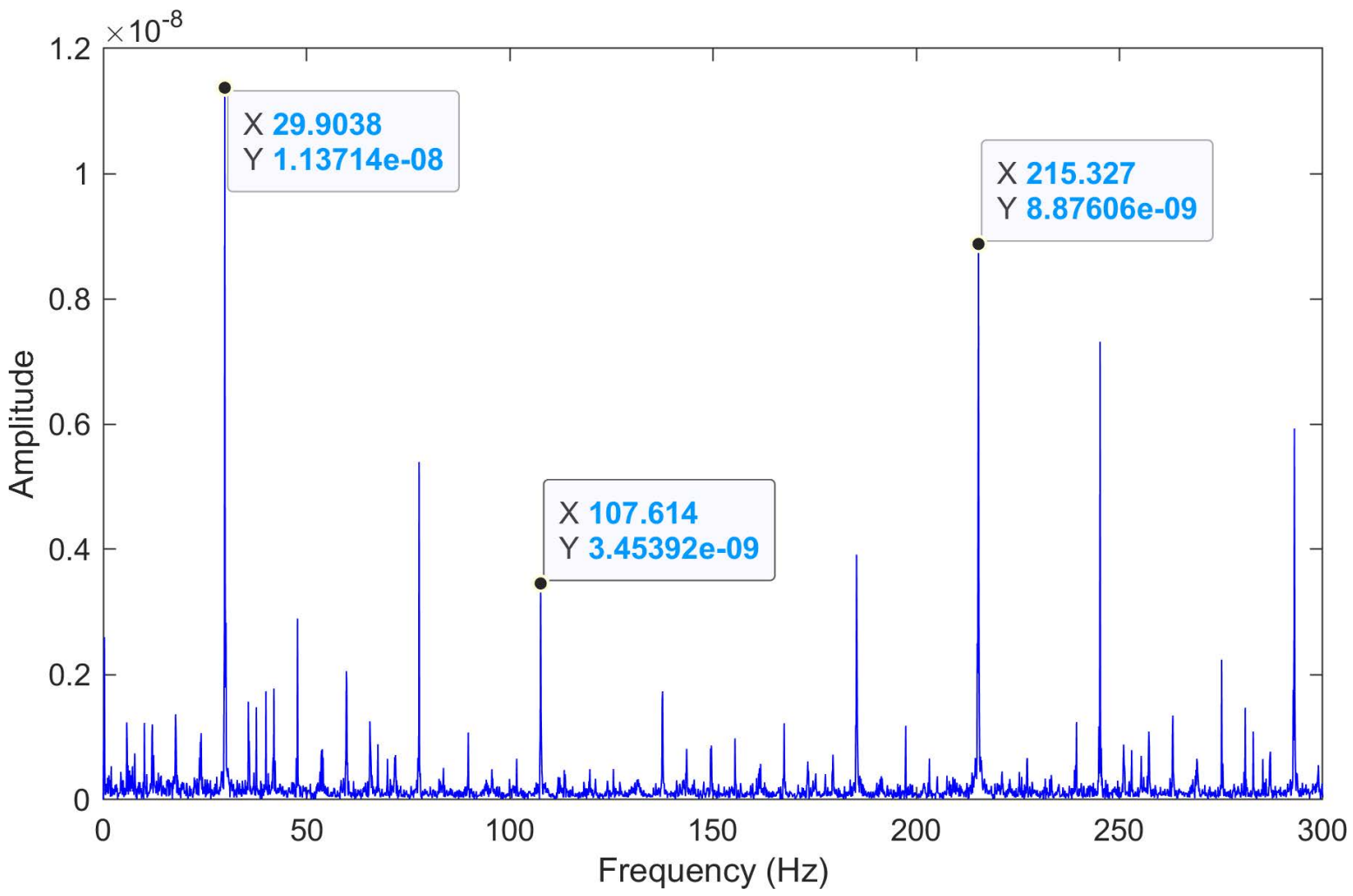


**Fig 22. The envelope spectrum after processing by the comparison method 2 of 130.mat.**


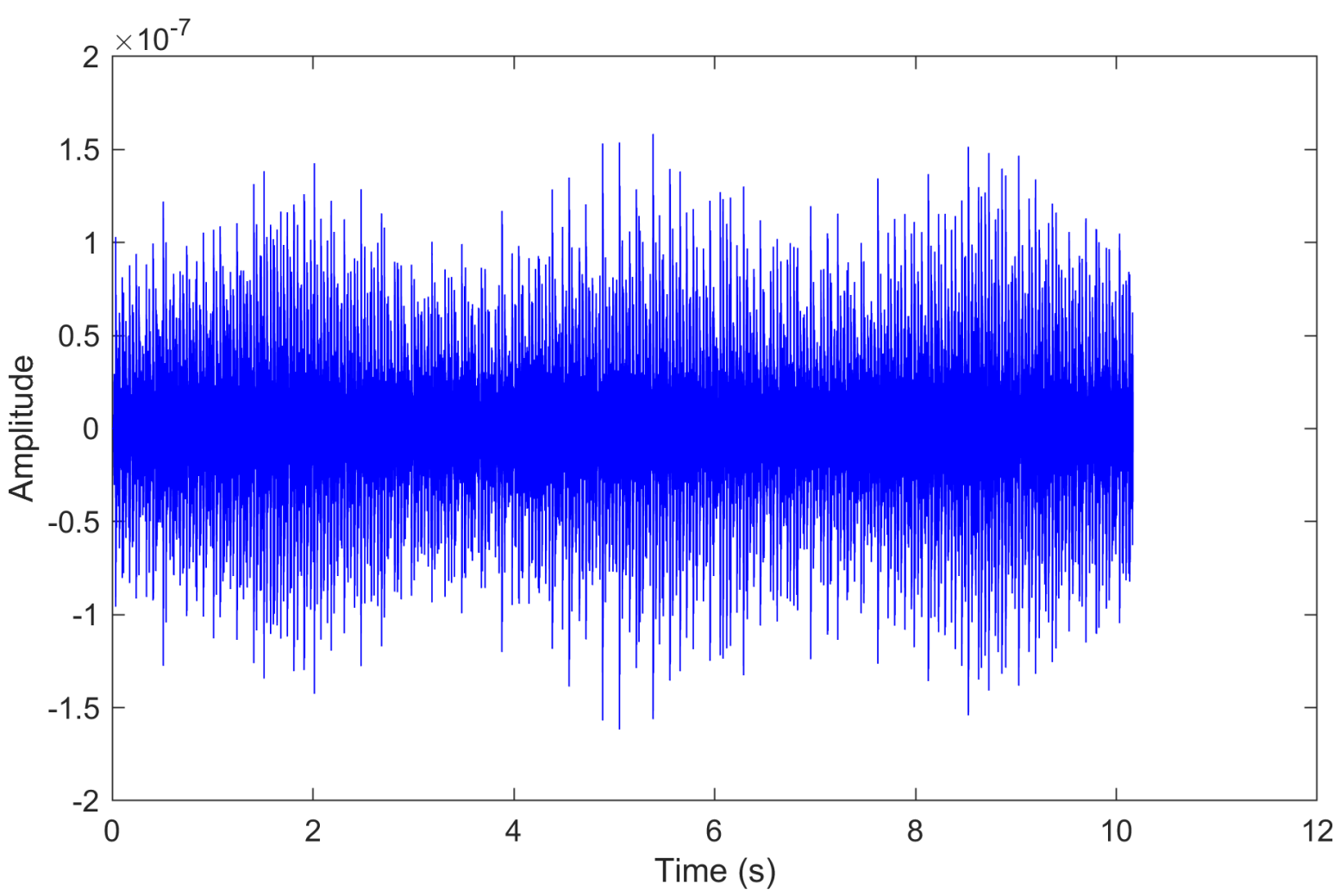


**Fig 23. The time-domain waveform after processing by the comparison method 2 of 130.mat.**


**Table 8. The experimental results of 130.mat.**

| \ | Theoretical value(Hz) | The proposed method(Hz) | Error rate(%) | Comparison method 1(Hz) | Error rate(%) | Comparison method 2(Hz) | Error rate(%) |
|---|---|---|---|---|---|---|---|
| $f_R$ | 29.95 | 29.904 | 0.154 | 29.297 | 2.229 | 29.904 | 0.154 |
| $f_O$ | 107.365 | 107.614 | 0.231 | 108.398 | 0.953 | 107.614 | 0.231 |
| $2f_O$ | 214.729 | 215.327 | 0.278 | 216.797 | 0.954 | 215.327 | 0.278 |


**Table 9. The data description of 236.mat.**

| Fault type | Fault Diameter | Motor Load | Motor Approximate Speed | Motor Rotation Frequency | Failure Frequency |
|---|---|---|---|---|---|
| Outer ring failure | 0.5334mm | 2 hp | 1750 r/min | 29.167 Hz | 104.566 Hz |


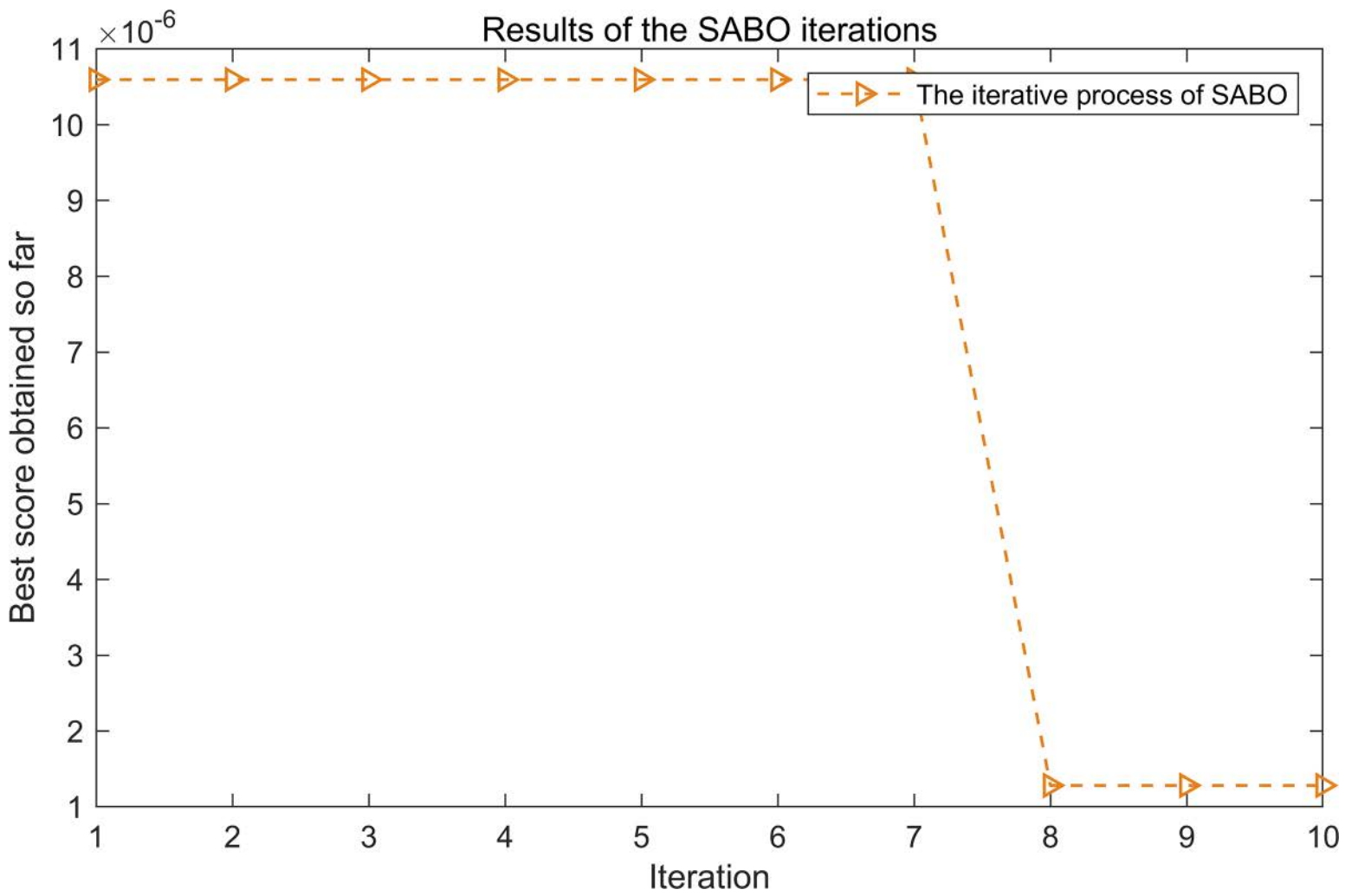


Fig 24. The iterative process of the SABO.


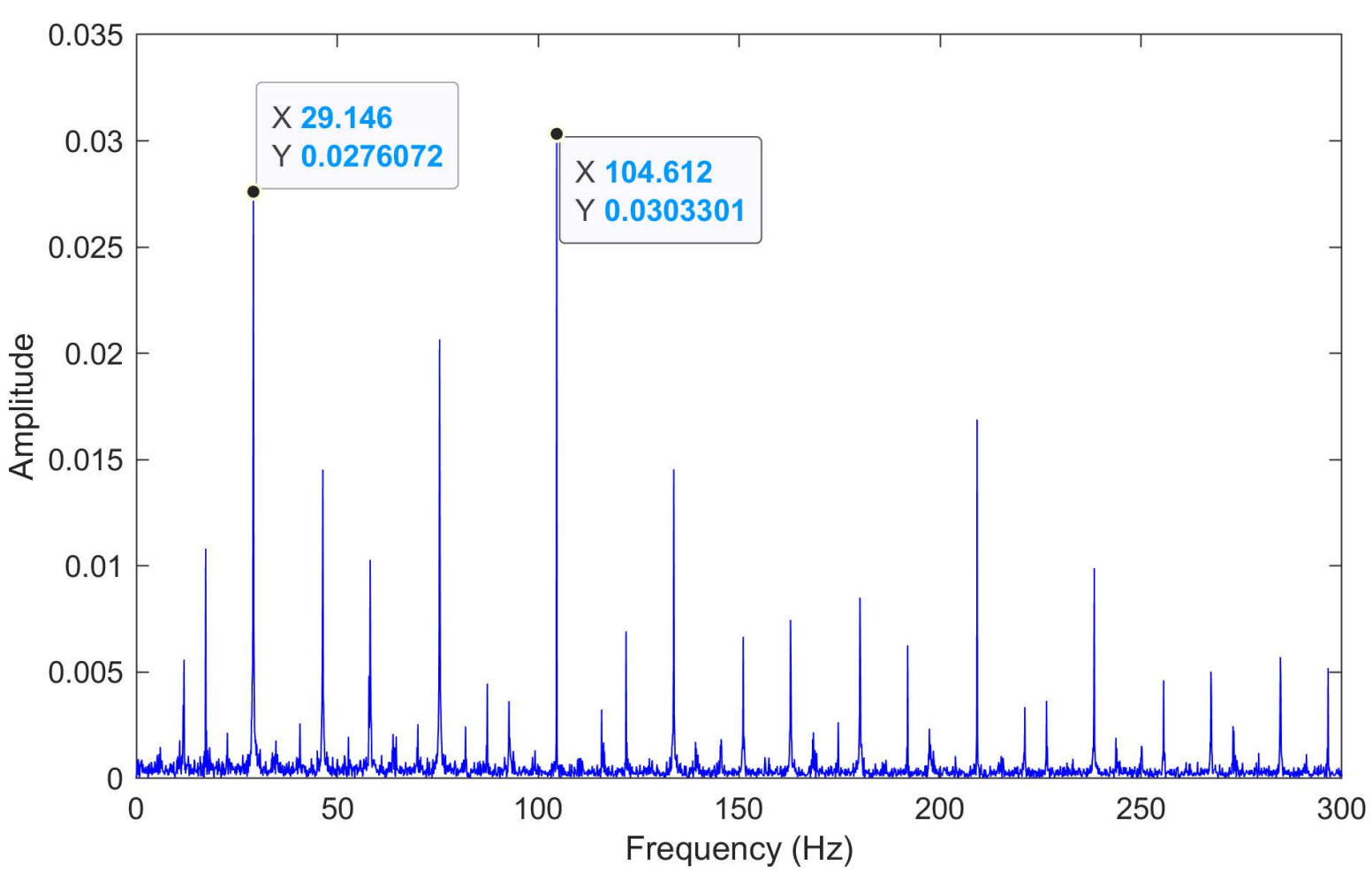


Fig 25. The envelope spectrum after processing by the proposed method of 236.mat.


The envelope spectrum of the comparison method 1 is shown in Fig 27 and its time-domain waveform is displayed in Fig 28. As shown in Fig 27, the horizontal coordinate of the highest peak is 29.2969 Hz, which matches the bearing's rotational frequency. The horizontal coordinate of the second-highest peak is 105.469 Hz, which is close to the theoretical outer ring fault frequency of the bearing.

The envelope spectrum of the comparison method 2 is shown in Fig 29 and its time-domain waveform is displayed in Fig 30. As shown in Fig 29, the horizontal coordinate of the highest peak is 29.146 Hz, which matches the bearing's rotational frequency. The horizontal coordinate of the other labeled peak is 104.612Hz, which is close to the theoretical outer ring fault frequency of the bearing.

**5.4.2 Discussion of experiment four.** The experimental results for the three methods are summarized in Table 10. By comparing the results of the three methods, two conclusions can be drawn. (1) Compared with the comparison method

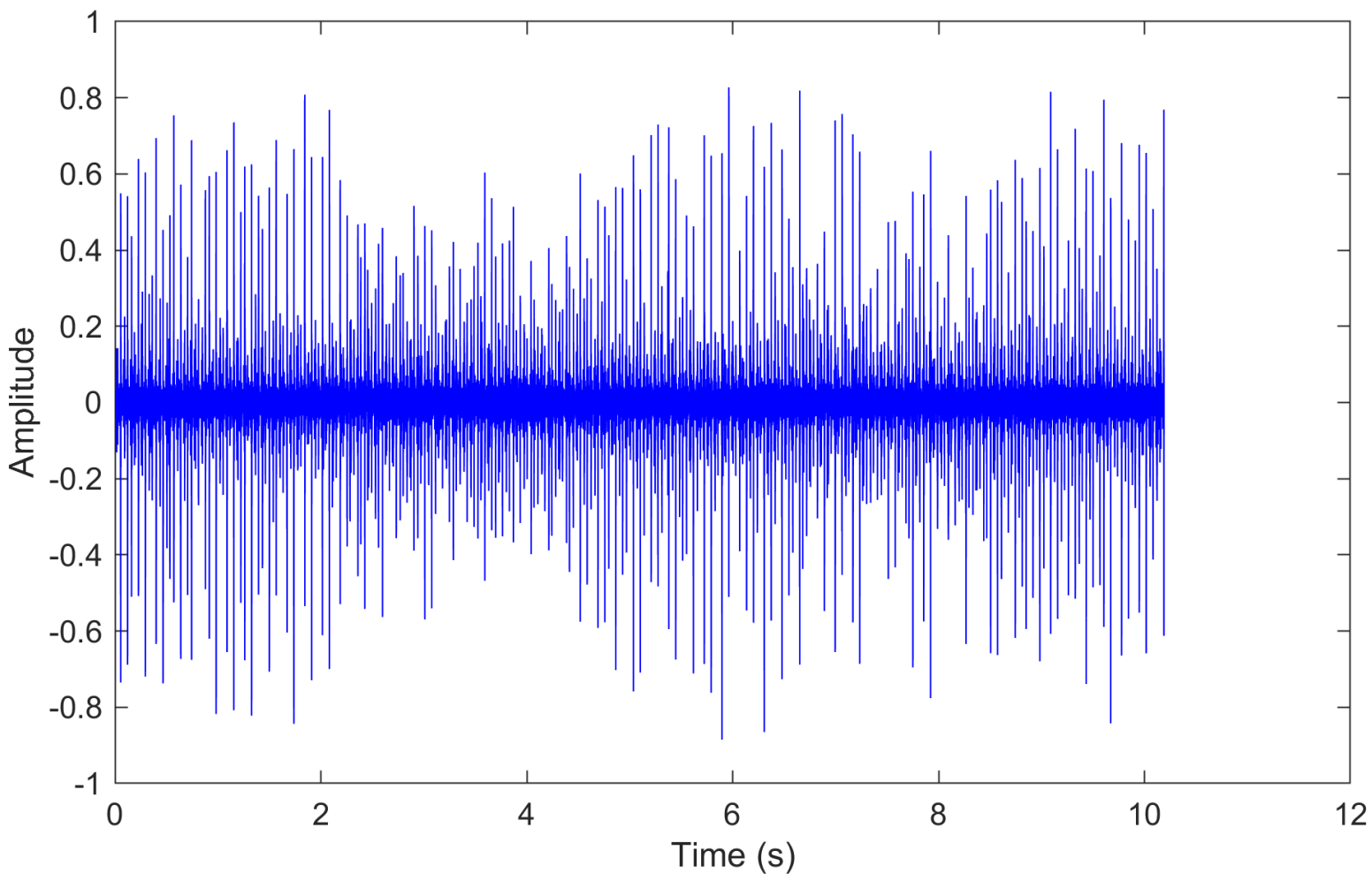


**Fig 26. The time-domain waveform after processing by the proposed method of 236.mat.**


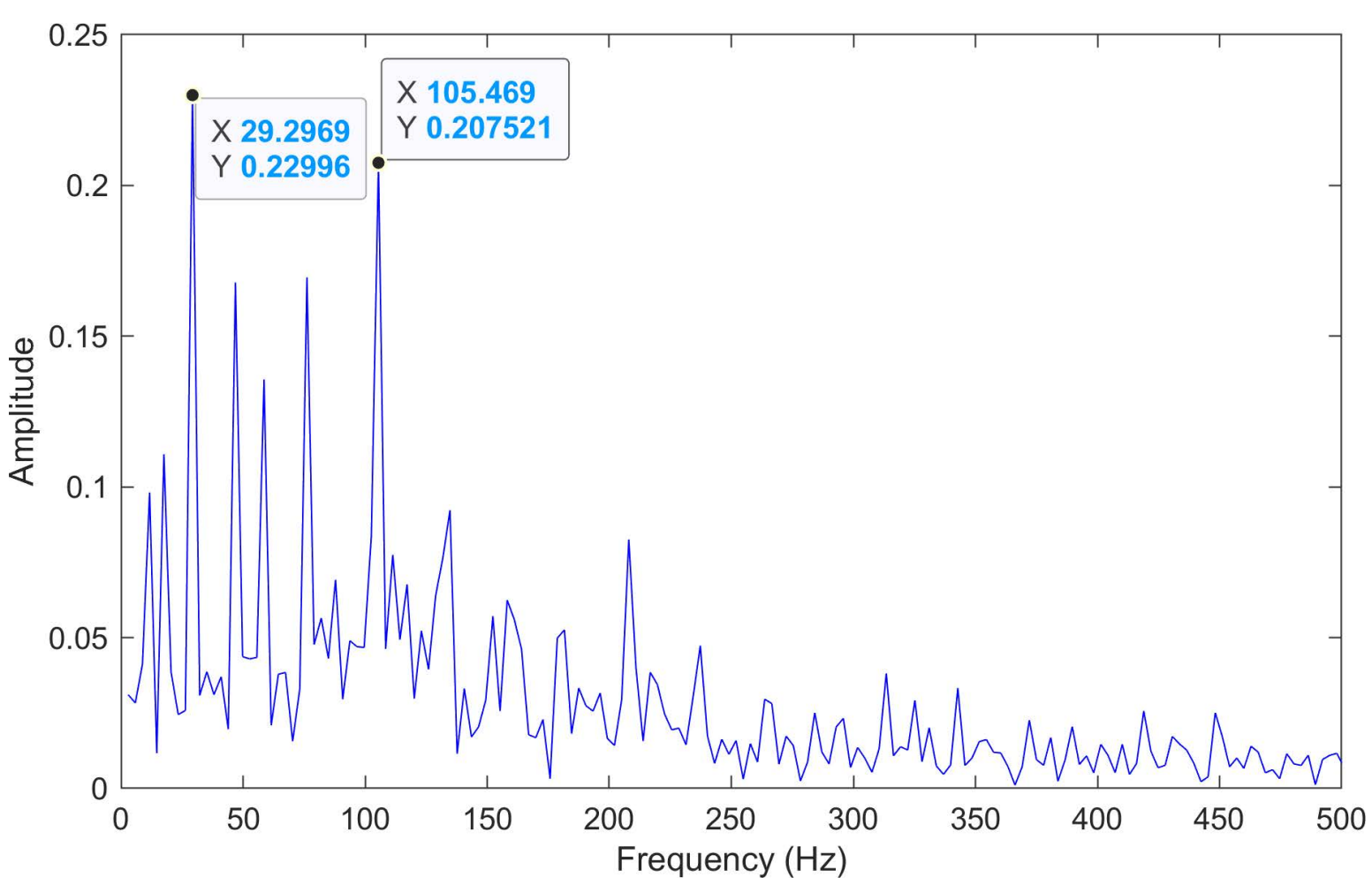


**Fig 27. The envelope spectrum after processing by the comparison method 1 of 236.mat.**


1, the proposed method obtains lower error rate of the information. (2) Although both the proposed method and the comparison method 2 identify the same fault frequencies, the fault information obtained by the proposed method is more pronounced.

## 6 Conclusions

This paper proposes a fault diagnosis method based on SABO and FMD. The method first employs SABO to optimize FMD parameters, then uses kurtosis values to screen IMFs containing fault information, and finally retains the envelope spectrum of the selected IMF to extract fault frequency information.

The main contributions of this paper are as follows:

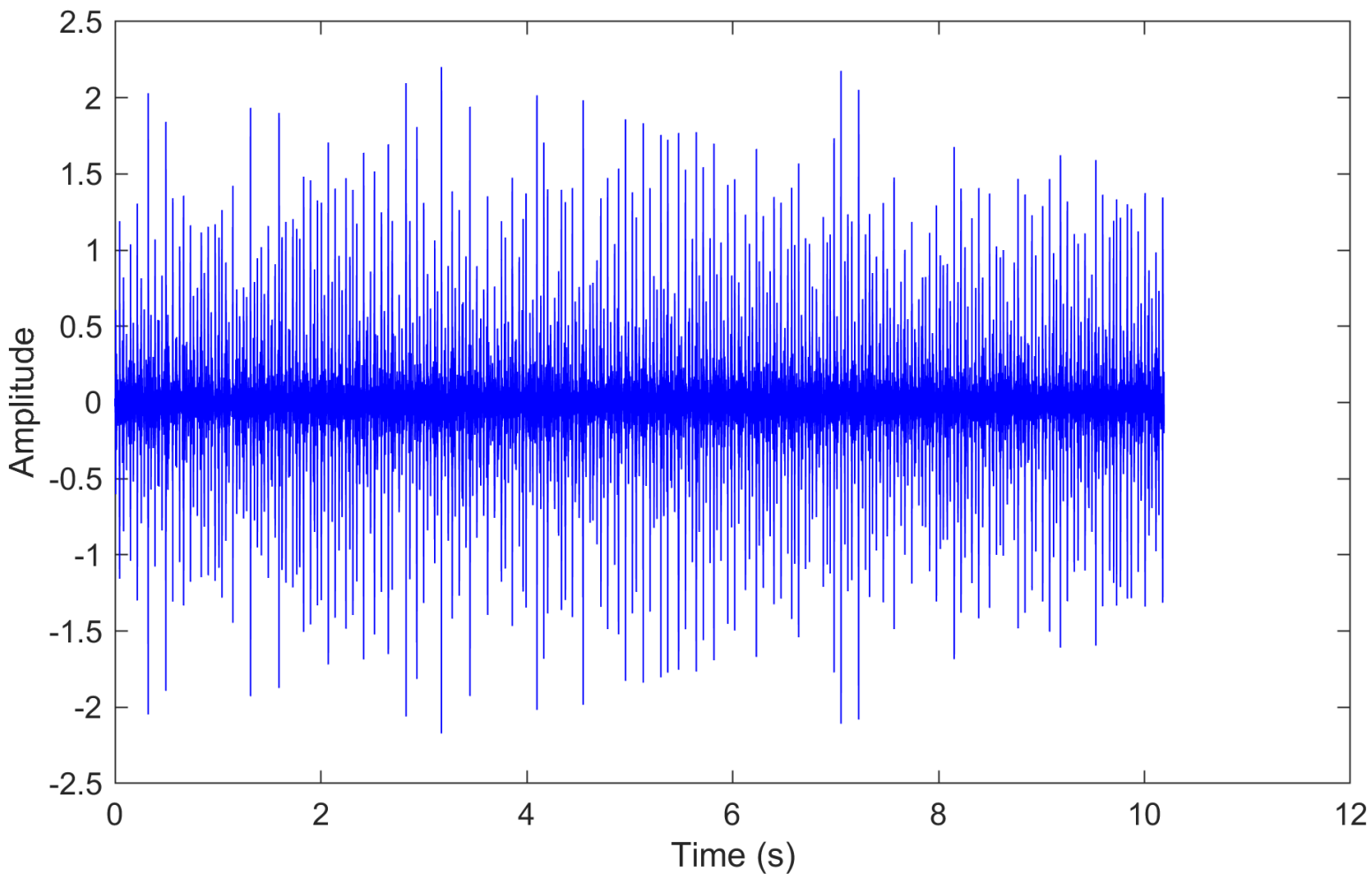


**Fig 28. The time-domain waveform after processing by the comparison method 1 of 236.mat.**


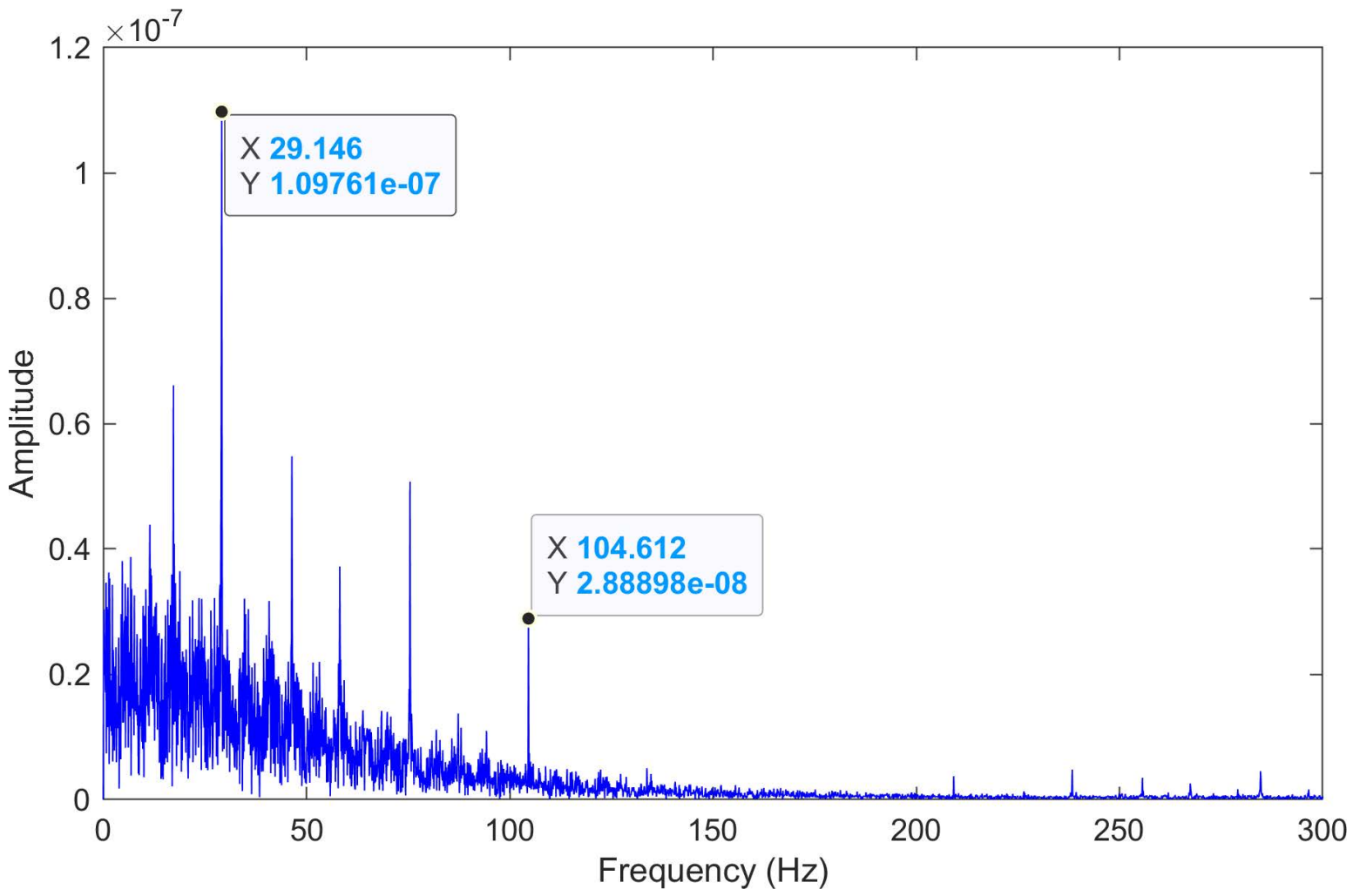


**Fig 29. The envelope spectrum after processing by the comparison method 2 of 236.mat.**


(1) The proposed method effectively addresses the issue of manual parameter selection in FMD decomposition, improving decomposition efficiency.

(2) The proposed method uses kurtosis values to effectively screen IMFs containing fault information, reducing computational costs and improving fault diagnosis efficiency.

(3) Compared with other similar algorithms, the proposed method offers a lower fault identification error rate and more distinct fault characteristics.

(4) The proposed method accurately and efficiently identifies fault information for different fault types, fault diameters, and motor speeds, demonstrating its potential for widespread application in fault diagnosis.

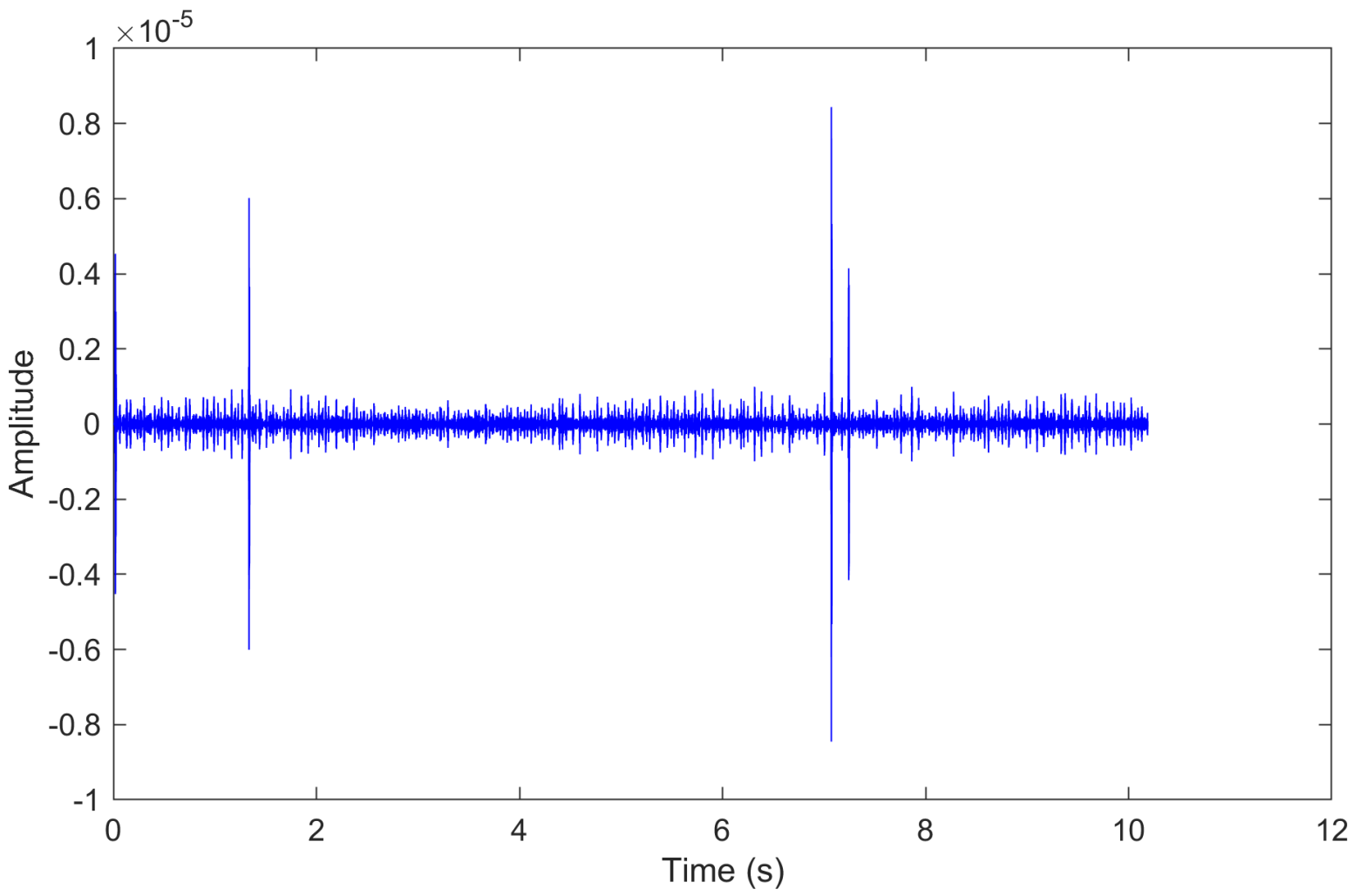


**Fig 30. The time-domain waveform after processing by the comparison method 2 of 236.mat.**


**Table 10. The experimental results of 236.mat.**

| \ | Theoretical value(Hz) | The proposed method(Hz) | Error rate(%) | Comparison method 1(Hz) | Error rate(%) | Comparison method 2(Hz) | Error rate(%) |
|---|---|---|---|---|---|---|---|
| $f_R$ | 29.167 | 29.146 | 0.072 | 29.297 | 0.444 | 29.146 | 0.072 |
| $f_O$ | 104.556 | 104.612 | 0.054 | 105.469 | 0.866 | 104.612 | 0.054 |


The method provides a novel approach to fault extraction, improving fault diagnosis accuracy and reducing computational costs. Future research will focus on developing more effective fault extraction and diagnosis methods for rolling bearings, as well as accurate classification techniques for extracted fault information.

## Supporting information

**S1 Data. This is the data used in the article and the results of the experiment.**
(ZIP)

## Author contributions

**Conceptualization:** Wei Xi.

**Data curation:** Wei Xi.

**Formal analysis:** Wei Xi, Jingjing Zhang.

**Funding acquisition:** Wei Xi, Fuyu Qiao.

**Investigation:** Wei Xi, Fuyu Qiao, Jingjing Zhang.

**Methodology:** Wei Xi, Fuyu Qiao.

**Project administration:** Fuyu Qiao, Jingjing Zhang.

**Software:** Wei Xi.

**Supervision:** Fuyu Qiao, Jingjing Zhang.

**Validation:** Fuyu Qiao, Jingjing Zhang.

**Visualization:** Jingjing Zhang.

**Writing – original draft:** Wei Xi.

**Writing – review & editing:** Wei Xi.

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
