## [Decision Letter · Decision Letter 0]

3 Jan 2025

Dear Dr. Zhang,

Thank you for submitting your manuscript to PLOS ONE. After careful consideration, we feel that it has merit but does not fully meet PLOS ONE’s publication criteria as it currently stands. Therefore, we invite you to submit a revised version of the manuscript that addresses the points raised during the review process.

ACADEMIC EDITOR: Please address all the comments using the separate response letter while a proof reading is highly recommended.

We look forward to receiving your revised manuscript.

Kind regards,

Qichun Zhang, PhD

Academic Editor

PLOS ONE

Journal Requirements:

[This research was funded by the Fundamental Research Funds for Central Universities (Grant No. 2023JCTD05)].

Please confirm at this time whether or not your submission contains all raw data required to replicate the results of your study. Authors must share the “minimal data set” for their submission. PLOS defines the minimal data set to consist of the data required to replicate all study findings reported in the article, as well as related metadata and methods (https://journals.plos.org/plosone/s/data-availability#loc-minimal-data-set-definition ).

If your submission does not contain these data, please either upload them as Supporting Information files or deposit them to a stable, public repository and provide us with the relevant URLs, DOIs, or accession numbers. For a list of recommended repositories, please see https://journals.plos.org/plosone/s/recommended-repositories .

Additional Editor Comments:

Based on the comments from the reviewers, the manuscript needs a major revision. Basically, the readability does not meet the criteria, please carefully proof read the contents using professional english writing. Diagram is needed to provide the clear idea of the proposed method while the comparative study is also essential. All typos should be corrected before resubmission along the format correction.

Reviewers' comments:

Reviewer's Responses to Questions

**Comments to the Author**

1. Is the manuscript technically sound, and do the data support the conclusions?

Reviewer #1: Partly

Reviewer #2: Yes

2. Has the statistical analysis been performed appropriately and rigorously?

Reviewer #1: Yes

Reviewer #2: Yes

3. Have the authors made all data underlying the findings in their manuscript fully available?

Reviewer #1: Yes

Reviewer #2: No

4. Is the manuscript presented in an intelligible fashion and written in standard English?

Reviewer #1: Yes

Reviewer #2: Yes

Reviewer #1: 1. The authors should cite more references to support the view.

2. The sections of the article should be numbered, as there is currently a lack of logical organization.

3. The article should provide an overall overview of the proposed method, preferably using a combination of images and text to provide a clear introduction.

4. There are many small errors in the article, including spelling and formatting. Additionally, the author should carefully check the formulas and the explanations of symbols within them, such as in Formula (7).

5. The photo of the experimental setup should be the raw image without any filters, and the hardware and parameter settings used in the experiment should be presented in one or two tables. The reviewer believes that there should be a certain level of progressive relationship from experiments 1 to 4, progressively demonstrating the superiority of the proposed method. The results from the experimental study should also provide the original vibration images. The images should be in vector format, and the text within them is too small and difficult to read. The analysis of the experimental results is not sufficiently in-depth.

6. The conclusion should be supported by relevant data, and it should also introduce the advantages and innovative aspects of the proposed methods.

7. The format of the references is confused.

Reviewer #2: In this paper, a fault diagnosis method that integrates Subtraction Average-Based Optimizer (SABO) with FMD has been proposed. The approach employs the index of minimum envelope entropy as a fitness function to derive an optimal combination of FMD parameters through the SABO optimization algorithm, subsequently facilitating fault diagnosis via envelope spectrum analysis. Experiments show the effectiveness. The paper's content structure is complete, but there are issues with non-standard formatting and poor readability. Here are my suggestions that may help the authors to improve this study.

a) The paper's format is not standardized enough, for example, there are deformations in the symbols and letters, the font in the figures is too small, and there are multiple blanks in the paper.

b) There is a lack of an overall scheme diagram, which reduces the readers' understanding of the methodological process of the paper.

c) The paper's methodology focuses on achieving accurate classification of rolling bearing faults from the perspective of data feature extraction, without discussing how to improve classification accuracy from the perspective of data feature processing. Is such a plan credible?

**Do you want your identity to be public for this peer review?** For information about this choice, including consent withdrawal, please see our Privacy Policy

Reviewer #1: No

Reviewer #2: No

---

## [Author Response · Author response to Decision Letter 0]

14 Feb 2025

We thank the reviewers/editors for their comments and suggestions on the article; the authors have revised the manuscript in accordance with the review comments and have submitted the responses as an attachment.

---

## [Decision Letter · Decision Letter 1]

28 Feb 2025

Dear Dr. Zhang,

Thank you for submitting your manuscript to PLOS ONE. After careful consideration, we feel that it has merit but does not fully meet PLOS ONE’s publication criteria as it currently stands. Therefore, we invite you to submit a revised version of the manuscript that addresses the points raised during the review process.

**ACADEMIC EDITOR: The reviewers returned a few additional comments which are important to improve the quality and readability. Please further revise the manuscript to address all these comments. Thus, a major revision is still needed. **

We look forward to receiving your revised manuscript.

Kind regards,

Qichun Zhang, PhD

Academic Editor

PLOS ONE

Reviewers' comments:

Reviewer's Responses to Questions

**Comments to the Author**

Reviewer #1: (No Response)

Reviewer #2: All comments have been addressed

2. Is the manuscript technically sound, and do the data support the conclusions?

Reviewer #1: (No Response)

Reviewer #2: Yes

3. Has the statistical analysis been performed appropriately and rigorously?

Reviewer #1: (No Response)

Reviewer #2: Yes

4. Have the authors made all data underlying the findings in their manuscript fully available?

Reviewer #1: (No Response)

Reviewer #2: Yes

5. Is the manuscript presented in an intelligible fashion and written in standard English?

Reviewer #1: (No Response)

Reviewer #2: Yes

Reviewer #1: After the first round of revisions, the reviewers acknowledged the authors' diligent efforts in modifying the manuscript; however, they noted that numerous issues remain unresolved.

1.The discussion of the newly cited references in the introduction appears discordant with the existing content.

2.The abstract does not carefully summarize the main points of the paper, especially lacking discussion of the proposed algorithm.

3.Several images retain bitmap formatting rather than adopting vector graphics standards.

4.The format of references is still confused.

Reviewer #2: (No Response)

**Do you want your identity to be public for this peer review?** For information about this choice, including consent withdrawal, please see our Privacy Policy

Reviewer #1: No

Reviewer #2: No

---

## [Decision Letter · Decision Letter 2]

30 Apr 2025

A Novel Rolling Bearing Fault Detect Method Based on Feature Mode Decomposition and Subtraction-Average-Based Optimizer

PONE-D-24-49078R2

Dear Dr. Zhang,

We’re pleased to inform you that your manuscript has been judged scientifically suitable for publication and will be formally accepted for publication once it meets all outstanding technical requirements.

Kind regards,

Qichun Zhang, PhD

Academic Editor

PLOS ONE

Reviewers' comments:

Reviewer's Responses to Questions

**Comments to the Author**

Reviewer #1: All comments have been addressed

Reviewer #2: All comments have been addressed

2. Is the manuscript technically sound, and do the data support the conclusions?

Reviewer #1: Yes

Reviewer #2: Yes

3. Has the statistical analysis been performed appropriately and rigorously?

Reviewer #1: Yes

Reviewer #2: (No Response)

4. Have the authors made all data underlying the findings in their manuscript fully available?

Reviewer #1: Yes

Reviewer #2: (No Response)

5. Is the manuscript presented in an intelligible fashion and written in standard English?

Reviewer #1: Yes

Reviewer #2: (No Response)

Reviewer #1: (No Response)

Reviewer #2: (No Response)

**Do you want your identity to be public for this peer review?** For information about this choice, including consent withdrawal, please see our Privacy Policy

Reviewer #1: No

Reviewer #2: No

---

## [Editor Report · Acceptance letter]

PONE-D-24-49078R2

PLOS ONE

Dear Dr. Zhang,

I'm pleased to inform you that your manuscript has been deemed suitable for publication in PLOS ONE. Congratulations! Your manuscript is now being handed over to our production team.

Kind regards,

on behalf of

Prof. Qichun Zhang

Academic Editor

PLOS ONE